# VisCoP: Visual Probing for Video Domain Adaptation of Vision Language Models

## Abstract

Large Vision Language Models (VLMs) excel at general visual reasoning tasks, but their performance degrades sharply when deployed in novel domains with substantial distribution shifts compared to what was seen during pretraining. Existing approaches to adapt VLMs to novel target domains rely on finetuning standard VLM components. Depending on which components are finetuned, these approaches either limit the VLMs ability to learn domain-specific features, or lead to catastrophic forgetting of pre-existing capabilities. To address this, we introduce **Vis**ion **Co**ntextualized **P**robing (**VisCoP**), which augments the VLM's vision encoder with a compact set of learnable *visual probes*, enabling domain-specific features to be learned with only minimal updates to the pretrained VLM components. We evaluate VisCoP across three challenging domain adaptation scenarios: cross-view (exocentric → egocentric), cross-modal (RGB → depth), and cross-task (human understanding → robot control). Our experiments demonstrate that VisCoP consistently outperforms existing domain adaptation strategies, achieving superior performance on the target domain, while better retaining capabilities from the source domain. We will release all code, models, and evaluation protocols to facilitate future research in VLM domain adaptation.

## 1 Introduction

Large Vision Language Models (VLMs) (OpenAI, 2025; Bai et al., 2025; Zhang et al., 2025; Xue et al., 2025) have achieved strong performance across a wide range of multi-modal understanding tasks, from open-ended video question answering (Zeng et al., 2023; Liu et al., 2023b) to complex spatial reasoning (Lai et al., 2023; Ranasinghe et al., 2024). Existing VLMs work by coupling Large Language Models (LLMs) (Qwen Team et al., 2025; Meta, 2024) together with pretrained vision encoders (Radford et al., 2021; Zhai et al., 2023) to enable powerful cross-modal reasoning capabilities. In practice, these models are primarily trained on large-scale, web-curated image/video-text corpora that cover broad but largely generic visual concepts (e.g., the human activities seen in internet videos) (Zhang et al., 2024; Chen et al., 2024b; Maaz et al., 2024; Rawal et al., 2024). As a result, when deployed in domains that differ significantly in viewpoint, sensing modality, or task structure, such as egocentric video understanding, depth-based perception, or robotic control, the performance of these VLMs degrade sharply due to distribution shift.

A common approach to bridge such distributional shift is to adapt a pretrained VLM to a target domain through finetuning on domain-specific video-QA instruction pairs. Unlike traditional video models (Bertasius et al., 2021; Arnab et al., 2021) that can solely focus on optimizing adaptation to a target domain, VLMs are expected to adapt *and* retain the general multi-modal capabilities learned during their pretraining. For example, consider a VLM pretrained on exocentric video understanding tasks that we wish to adapt to tasks recorded from the egocentric viewpoint. After adaptation, the model should still retain its performance on tasks recorded from the exocentric viewpoint.

Existing approaches for domain adaptation in VLMs follow multi-stage training schemes (Li et al., 2023a) in which different components are trained in each stage. Training only lightweight components, such as the vision-language connector, retains pretrained knowledge but limits domain-specific visual understanding. In contrast, training the vision encoder enables specialized visual understanding, albeit at the cost of catastrophic forgetting of pretrained knowledge (Yang et al., 2023; Zang et al., 2024; Li et al., 2024). However, when the dominant shift between the pretraining and target domains is *visual*, as is the case in many video settings (e.g., exocentric → egocentric viewpoint,

RGB → depth modality, visual perception → robotic control), learning domain-specific visual representation is necessary. This raises the fundamental question: *how can VLMs be adapted to novel domains to learn domain-specific visual features, without requiring updates to its visual encoder?*

To this end, we introduce **Vi**sion **Co**ntextualized **P**robing, dubbed VISCOP, a mechanism that enables adaptation of pretrained VLMs to a novel target domain, while retaining its general-purpose visual representations learned during pretraining. VISCOP probes a frozen vision encoder via a compact set of learnable tokens that form an alternative adaptation pathway for extracting domain-specific visual signals. Motivated by the progressive emergence of semantics across transformer depths (Vaswani et al., 2017; Bertasius et al., 2021; Liu et al., 2021), the visual probes interact layerwise with intermediate representations of the frozen visual encoder. This design enables the probes to capture domain-specific patterns at multiple levels of abstraction, which can be fed to the LLM to enhance domain-specific visual reasoning. Unlike methods (Li et al., 2023b; Alayrac et al., 2022; Ha et al., 2024; Ryoo et al., 2025) that only leverage the high-level representations from the final layer of the VLM's visual encoder, our multi-layer probing is able to extract representations from earlier layers and propagate them forward, surfacing domain-relevant cues that might have otherwise been dis-

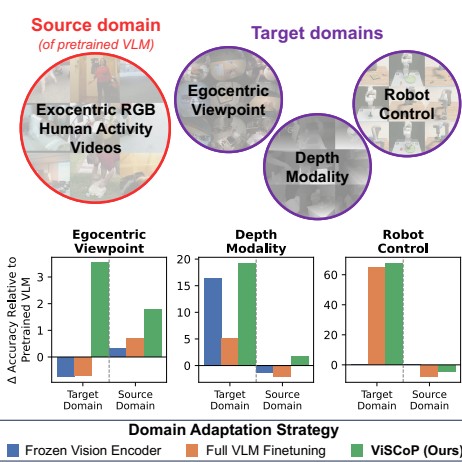

Figure 1: **Domain adaptation performance of different adaptation strategies.** VISCOP achieves superior target domain performance while better retaining source domain knowledge compared to other strategies.

carded by the frozen vision encoder. Empirically, we find that the representations learned via the VISCOP adaptation pathway enable effective cross-view, cross-modal, and cross-task adaptation of VLMs, while retaining their broad capabilities learned during pretraining. Metaphorically, the name VISCOP reflects its role as a "*traffic cop*", directing gradient flows away from the visual encoder and towards an alternative pathway for learning domain-specific visual features, avoiding the "*crash*" (catastrophic forgetting) that would otherwise occur if gradients flowed through the visual encoder.

To summarize, our contributions:

1. We propose VISCOP (**Vis**ion **Co**ntextualized **P**robing), a novel domain adaptation strategy for VLMs that learns domain-specific visual representations through probing of a frozen vision encoder, enabling effective domain transfer and preventing catastrophic forgetting of multi-modal capabilities learned during pretraining.

2. We establish a comprehensive evaluation setting for domain adaptation in VLMs, spanning three challenging target domains: cross-view (exocentric → egocentric), cross-modality (RGB → depth), and cross-task (action understanding → robotic control), along with standardized metrics to evaluate performance. We will release code and data to facilitate future research on domain adaptation in VLMs.

3. Our experiments show that post-adaptation, VLMs trained with VISCOP outperform alternative domain adaptation strategies across diverse target domains, while retaining more knowledge of the source domain, as illustrated in Figure 1.

## 2 RELATED WORKS

**Domain adaptation in vision-language encoders.** Domain adaptation of contrastively trained vision-language encoders, such as CLIP (Radford et al., 2021; Rasheed et al., 2023), is typically achieved through prompt tuning or adapter-based approaches. Both strategies aim to learn domain-specific features while keeping the pretrained vision and text encoders frozen. To accomplish this, prompt tuning approaches (Zhou et al., 2022; Zhu et al., 2023; Yao et al., 2023) introduce learnable prompt vectors as additional input to the text encoder, steering the model toward target domain. Adapter-based approaches (Yang et al., 2023; Gao et al., 2023) insert lightweight trainable modules directly into the encoder space, thus updating their pretrained representations. In contrast to these

approaches, VISCOP addresses the setting of domain adaptation in generative VLMs, enabling them to learn domain-specific features without requiring updates to the pretrained encoder representations.

**Domain adaptation in VLMs.** Domain adaptation in VLMs has largely been achieved through data-centric strategies rather than through architectural changes (Cheng et al., 2025). Existing approaches typically leverage automated pipelines (Mohbat & Zaki, 2024; Reilly et al., 2025a) or closed-source VLMs (Li et al., 2023a; Chen et al., 2024a) to curate visual-instruction pairs from existing datasets in the target domain. Their adaptation strategy usually follows a multi-stage training scheme similar to LLaVA (Liu et al., 2023a), where different VLM components are selectively trained at each stage. However, the choice of trainable components creates a trade-off between extracting domain-specific features and retaining pretrained knowledge. Training only lightweight connectors retains pretrained knowledge but limits domain-specific visual understanding, while training the vision encoder enables specialized visual understanding at the cost of catastrophic forgetting. VISCOP avoids this trade-off through the introduction of visual probes that extract domain-specific features from a frozen vision encoder, enabling adaptation without disrupting the pretrained visual representations.

**Visual probing vs. visual compression.** Several approaches employ learnable tokens to bridge vision and language modalities (Ha et al., 2024; Ryoo et al., 2025; Zohar et al., 2025) through architectures leveraging the Q-Former and Perceiver Resampler modules. Q-Former (Li et al., 2023b) leverages learnable queries that cross-attend to representations from the final layer of the vision encoder, aggregating visual information into a reduced set of tokens for computational efficiency. Perceiver Resampler (Alayrac et al., 2022) operates similarly, aiming to compress the visual representations into a fixed number of learnable tokens. The visual probes proposed in VISCOP differ fundamentally, as they are designed to *extract* novel domain-specific visual representations rather than to simply *compress* pretrained ones. This is enabled by their interaction with intermediate representations of the vision encoder, allowing the probes to extract domain-specific representations that are not propagated to the final representation of the pretrained vision encoder (Radford et al., 2021; Zhai et al., 2023).

## 3 PROBLEM FORMULATION

Let $\mathcal{S}$ denote the *source domain*, on which the vision-language model $f_{\theta^0}$ has been pretrained, and let $\mathcal{T}$ denote the *target domain*, the domain of interest for adaptation. The two domains differ in their underlying distributions (e.g., viewpoint, modality, or task), which causes $f_{\theta^0}$ to perform poorly when directly applied to $\mathcal{T}$.

Training supervision in these domains is provided as video-QA pairs $(v, q, a)$, where $v$ is a video, $q$ is an instruction or question, and $a$ is the corresponding response. While $f_{\theta^0}$ has been pretrained on samples $(v, q, a) \sim \mathcal{S}$, at adaptation time we only assume availability of target domain samples $(v, q, a) \sim \mathcal{T}$. The objective of domain adaptation is to update the pretrained parameters $\theta^0$ to obtain $\theta^\star$ that improves performance on domain $\mathcal{T}$, while retaining performance on domain $\mathcal{S}$. Formally,

$$R_{\mathcal{T}}(\theta^\star) < R_{\mathcal{T}}(\theta^0) \quad \text{and} \quad R_{\mathcal{S}}(\theta^\star) \approx R_{\mathcal{S}}(\theta^0)$$

where $R_{\mathcal{D}}$ denotes the VLM's expected autoregressive next-token prediction loss under domain $\mathcal{D}$. In summary, our problem statement considers adaptation of a pretrained VLM to a novel domain using only video-QA pairs from that domain. The objective is to improve target-domain performance while minimizing catastrophic forgetting of source-domain capabilities. In the next section, we introduce our proposed method, which enables balanced domain adaptation under these constraints.

## 4 METHOD: VIDEO DOMAIN-ADAPTIVE VLM

Given a video input $\mathbf{V} = \{\boldsymbol{I}_t\}_{t=1}^{T}$ consisting of $T$ frames, the goal of the VLM is to generate the response corresponding to the input instruction in an autoregressive manner.

### 4.1 PRELIMINARY

Existing VLMs for video representation learning (Zhang et al., 2025; Bai et al., 2025) consist of three standard components: **(i)** a vision encoder that maps visual inputs into a sequence of spatio-temporal tokens, **(ii)** a vision-language connector that projects the visual tokens to the embedding space of a language model, and **(iii)** an LLM that processes the projected visual tokens jointly with language tokens to enable multi-modal reasoning. For the input video $\mathbf{V}$, each frame $\boldsymbol{I}_t$ is processed independently by the vision encoder through a stack of $L$ transformer layers. The visual tokens after

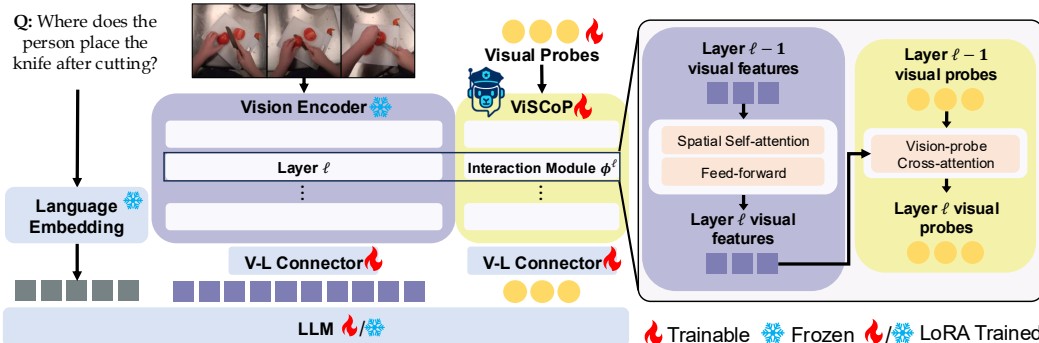

Figure 2: **Architecture of our proposed VISCOP.** Learnable visual probes are conditioned on intermediate representations of a frozen vision encoder through vision-probe cross-attention, which extracts domain-specific features that may have otherwise been discarded by the frozen encoder.

the $\ell$-th layer are denoted as

$$\mathbf{X}_t^\ell \in \mathbb{R}^{N \times d_v}, \quad \ell = 1, \dots, L$$

where $N$ is the number of spatial patch tokens per frame and $d_v$ is the embedding dimension of the vision encoder. Concatenating these tokens over time yields $\mathbf{X}^\ell \in \mathbb{R}^{(TN) \times d_v}$ which represents the sequence of spatio-temporal visual tokens at the $\ell$-th layer of the vision encoder. The final layer outputs $\mathbf{X}^L$ are then projected to the language embedding space via a vision-language connector $\mathcal{C}$ to obtain the visual embeddings used as input to the LLM

$$\mathbf{E} = \mathcal{C}(\mathbf{X}^L) \in \mathbb{R}^{(T\tilde{N}) \times d_{\mathrm{lm}}}$$

where $\tilde{N}$ is the number of visual tokens input to the LLM after spatial downsampling (Zhang et al., 2025). and $d_{\mathrm{lm}}$ is the embedding dimension of the LLM.

The VLM is then trained to optimize a standard autoregressive next token prediction loss. Specifically, given the visual embeddings $\mathbf{E}$ and the tokenized QA pair $(\mathbf{Q}, \mathbf{A})$, we optimize the likelihood of predicting $\mathbf{A}$ conditioned on the visual embeddings and the question

$$P(\mathbf{A} \mid \mathbf{E}, \mathbf{Q}) = \prod_{j=1}^{\mathrm{Len}} P_{\boldsymbol{\theta}}(\mathbf{a}_j \mid \mathbf{E}, \mathbf{Q}, \mathbf{A}_{<j})$$

where $\boldsymbol{\theta}$ are the trainable parameters of the VLM, Len indicates the token length of $\mathbf{A}$, and $\mathbf{A}_{<j}$ represents the subsequence of answer tokens preceding position $j$.

For domain-adaptive post training of VLMs, finetuning the vision encoder of a pretrained VLM for a target domain $\mathcal{T}$ often leads to overfitting on $\mathcal{T}$ and catastrophic forgetting of the source domain (Yang et al., 2023; Zang et al., 2024; Li et al., 2024). To mitigate this trade-off, a domain-adaptive pathway is required that adapts the VLM to $\mathcal{T}$ while retaining performance on $\mathcal{S}$.

### 4.2 VISCOP: VISION CONTEXTUALIZED PROBING

To capture the relevant visual context that would otherwise be lost by freezing the vision encoder, we propose **Vis**ion **Co**ntextualized **P**robing (**VISCOP**), a mechanism that augments the vision encoder with a compact set of learnable tokens, called *visual probes*, and an interaction module that acts as a semantic interface between the probes and intermediate visual representations, as illustrated in Figure 2. In this section, we introduce how domain-adaptive VLMs are trained with VISCOP.

VISCOP augments the frozen vision encoder of a VLM with a compact set of $M$ learnable *visual probes* $\mathbf{P} \in \mathbb{R}^{M \times d_v}$. The probes are trained to extract domain-specific spatio-temporal cues from intermediate representations of the vision encoder. To enable this extraction, a learnable *interaction module* $\Phi^\ell$ inserted at each layer of the vision encoder conditions the probes on the hierarchical representations of the vision encoder at layer $\ell$:

$$\mathbf{P}^{\ell+1} = \Phi^\ell(\mathbf{P}^\ell, \mathbf{X}^\ell).$$

Concretely, $\Phi^\ell$ is implemented as a vision-probe cross-attention between the visual embeddings and the probes at layer $\ell$. Let $(\boldsymbol{W_q}, \boldsymbol{W_k}, \boldsymbol{W_v})$ be the projection matrices in $\Phi^\ell$, then the probe update is

$$\mathbf{P}^\ell \;=\; \mathrm{softmax}\left( \frac{\mathbf{P}^\ell \boldsymbol{W_q^\ell}(\mathbf{X}^\ell \boldsymbol{W_k^\ell})^\top}{\sqrt{d_v}} \right) (\mathbf{X}^\ell \boldsymbol{W_v^\ell}),$$

Each $\Phi^\ell$ is parameterized independently, enabling layer-specific aggregation of low- to high-level visual semantics. E3: In contrast to the vision encoder of existing VLMs, which only learns spatial relationships through intra-frame self-attention, the visual probes attend to *all* frames in the video, enabling them to learn complex spatio-temporal relationships. In some settings, such as robotic control, vision-probe cross-attentions are restricted to spatial tokens only.

After the final layer, the updated probes $\mathbf{P}^L$ are projected to the language embedding space via a dedicated connector $\mathcal{C}_{\mathrm{probe}}$, $$\mathbf{Z} = \mathcal{C}_{\mathrm{probe}}(\mathbf{P}^L) \in \mathbb{R}^{M \times d_{\mathrm{lm}}},$$

and the VLM is trained with the standard autoregressive objective additionally conditioned on $\mathbf{Z}$:

$$P(\mathbf{A} \mid \mathbf{E}, \mathbf{Q}, \mathbf{Z}) = \prod_{j=1}^{\mathrm{Len}} P_{\boldsymbol{\theta}}(\mathbf{a}_j \mid \mathbf{E}, \mathbf{Q}, \mathbf{Z}, \mathbf{A}_{<j}).$$

Thus, the probes act as low-dimensional control knobs that bias learning toward domain-relevant structure and away from spurious artifacts. This is reinforced by applying updates through the probe connector, and through LoRA (Hu et al., 2021) updates in the LLM embedding space, which confine parameter changes to a low-rank, probe-defined visual subspace that preserves generalizable behavior while enabling targeted specialization.

## 5 EXPERIMENTS

We evaluate VISCOP for effective domain adaptation and minimal forgetting. Section 5.1 details the setup (architecture, training, metrics); Section 5.2 reports results on egocentric, depth, and robotic-control targets; Section 5.3 presents ablations and representation analyses of the probes and interaction modules.

### 5.1 EXPERIMENTAL SETTING

**VLM Architecture.** We consider a VLM architecture consisting of a SigLIP (Zhai et al., 2023) vision encoder, Qwen 2.5 (Qwen Team et al., 2025) LLM, and a 2-layer MLP vision-language connector, with all modules initialized from the pretrained weights of VideoLLaMA3 (Zhang et al., 2025). The embedding dimension of the vision encoder is $d_v = 1152$, and the embedding dimension of the LLM is $d_{\mathrm{lm}} = 3584$. We refer to this pretrained model as the *base* VLM, and to models adapted to a target domain as *expert* VLMs. To adapt the base VLM to a target domain, we perform finetuning on the target domain with a learning rate of $1 \times 10^{-5}$ for the LLM and vision-language connector, and a learning rate of $2 \times 10^{-6}$ for the vision encoder (when trainable). The model is finetuned on 4 NVIDIA H200 GPUs for 3 epochs when adapting to video domains, or 2 epochs when adapting to robotic control domains.

**VISCOP Details.** By default, VISCOP operates at every layer of the vision encoder and employs $M = 16$ visual probes unless otherwise stated. The visual probes are initialized from the normal distribution $\mathcal{N}(0, 0.02)$. Each interaction module $\Phi^\ell$ is implemented as a multi-head cross-attention (Vaswani et al., 2017), and its weights are initialized from the self-attention weights of the vision encoder at layer $\ell$. During domain adaptation, we freeze the vision encoder and update only the visual probes, interaction modules, vision–language connectors, and the LLM's LoRA parameters. For adaptation to video understanding domains, we update the LLM using LoRA ($r = 16$), while the entire LLM is updated when adapting to the robotic control domain.

**Adaptation Metrics.** We evaluate the domain adaptation of VLMs across two dimensions: **(i)** their "*improvement*" on the target domain $\mathcal{T}$, and **(ii)** their "*retention*" on the source domain $\mathcal{S}$. Improvement on the target domain is measured as the performance difference between the expert and base VLMs on target domain benchmarks; retention is the corresponding difference on source domain benchmarks. If $\mathrm{Acc}_{\mathcal{D}}$ denotes the average accuracy over all benchmarks within the domain $\mathcal{D}$, then the metrics are computed by:

$$\Delta_{\mathrm{target}} = \mathrm{Acc}_{\mathrm{target}}^{\mathrm{expert}} - \mathrm{Acc}_{\mathrm{target}}^{\mathrm{base}} \qquad\qquad \Delta_{\mathrm{source}} = \mathrm{Acc}_{\mathrm{source}}^{\mathrm{expert}} - \mathrm{Acc}_{\mathrm{source}}^{\mathrm{base}}$$

## 5.2 Source and Target Domains

The source domain $\mathcal{S}$ is fixed throughout this paper: exocentric RGB videos of human actions reflecting the samples used to train generic VLMs for video representation learning. Our target domains $\mathcal{T}$ deliberately shift the input distribution, and consist of (1) egocentric video understanding, (2) depth-modality video understanding, and (3) robotic control. **E6:** All data (videos and instructions) in our chosen target domain benchmarks were not used in the pretraining of the base VLM (Zhang et al., 2025). We evaluate VISCoP's adaptation to each target while measuring retention of source domain competencies: (i) when adapting to egocentric video, exocentric understanding should be preserved; (ii) when adapting to depth video, RGB understanding should be preserved; and (iii) when adapting to robotic control, human-action understanding should be preserved.

**Training datasets.** For ego and depth video understanding domains, we adapt using EgoExo4D (Grauman et al., 2025), a large-scale multi-view dataset containing time-synchronized egocentric and exocentric videos of skilled human activities. We utilize a total of 24,688 videos from the keystep recognition subset to generate 74,064 video instruction pairs. These instructions are recaptioned from the instruction pairs provided in Reilly et al. (2025b). For the **egocentric** target domain, we adapt on 45,888 egocentric video-instruction pairs. For the **depth** target domain, we convert all exocentric RGB videos to depth using DepthAnythingV2 (Yang et al., 2024) while keeping the language instructions unchanged, yielding 28,176 depth instruction pairs.

We perform adaptation to the **robotic control** domain in both simulated and real-world robot environments. In the *simulated environment*, we leverage the training set of VIMA-Bench (Jiang et al., 2023). VIMA-Bench contains 17 object manipulation tasks with an action space comprising two 2D coordinates (for pick and place positions) and two quaternions (for rotation). Since the training set of VIMA-Bench lacks natural language instructions by default, we leverage the instruction pairs generated in LLaRA (Li et al., 2025), resulting in 13,922 instruction pairs across 7,995 action trajectories. In the *real-world environment*, we collect a dataset using a 6-DoF xArm 7 robot arm deployed in a tabletop manipulation setting. This dataset, which we refer to as xArm-Det, contains 1,007 instruction pairs depicting novel objects and spatial configurations not present in simulation. During adaptation, we train jointly on VIMA-Bench and xArm-Det, resulting in a total of 14,929 instruction pairs. The large-scale simulated data enables the model to learn manipulation skills, while xArm-Det exposes the model to our novel robot environment. Illustrations of our real-world robot environment and examples from VIMA-Bench are provided in Appendix A.1.

Table 1: **Egocentric Video Understanding Experts.** Performance of adaptation strategies on the egocentric target domain and exocentric source domain. Adaptation strategy correspond to the trainable components of the VLM: **VL-C** = Vision Language Connector, **VE** = Vision Encoder, and **LLM** = Large Language Model. $\Delta_{\text{target}}$ and $\Delta_{\text{source}}$ denote relative gains over the Base VLM.

| Adaptation Strategy | | | Egocentric Benchmarks (Target) | | | | | | Exocentric Benchmarks (Source) | | | | | Adaptation Metrics | |
| --- | --- | --- | --- | --- | --- | --- | --- | --- | --- | --- | --- | --- | --- | --- | --- |
| | | | *Ego-in-Exo PerceptionMCQ (Ego RGB)* | | | | | | | | | | | $\Delta_{\text{target}}$ | $\Delta_{\text{source}}$ |
| VL-C | VE | LLM | Action Und. | Task Regions | HOI | Hand Ident. | EgoSchema | Avg | NeXTQA | VideoMME | ADL-X MCQ | ADL-X Desc | Avg | ($\uparrow$) | ($\uparrow$) |
| Base VLM | | | 75.37 | 74.88 | 75.56 | 65.38 | 60.98 | 70.43 | **84.32** | **65.37** | 77.36 | 70.65 | 74.42 | - | - |
| ✓ | ✗ | ✗ | 73.00 | 76.71 | 72.85 | 65.51 | 60.43 | 69.70 | 84.21 | 62.67 | 76.56 | 75.51 | 74.74 | -0.74 | +0.31 |
| ✓ | ✓ | ✗ | 76.13 | **82.93** | 73.32 | 64.86 | 61.14 | 71.68 | 83.87 | 61.41 | 77.05 | 76.09 | 74.61 | +1.24 | +0.18 |
| ✓ | ✗ | ✓ | 73.28 | 82.68 | 72.96 | **65.77** | 60.31 | 71.00 | 82.34 | 64.26 | 78.21 | 70.89 | 73.93 | +0.57 | -0.50 |
| ✓ | ✗ | LoRA | 73.49 | 74.27 | 74.50 | 64.99 | 61.52 | 69.75 | 84.24 | 64.41 | 77.42 | 74.36 | 75.11 | -0.68 | +0.68 |
| ✓ | VISCoP | LoRA | **81.28** | 82.80 | **78.75** | 64.86 | **62.11** | **73.96** | 84.31 | 64.70 | **78.97** | **76.78** | **76.19** | **+3.53** | **+1.77** |

### 5.2.1 Egocentric Video Understanding

**Target and source benchmarks.** For evaluation on the **target domain**, we evaluate on the Ego-in-Exo PerceptionMCQ (Reilly et al., 2025b) and EgoSchema (Mangalam et al., 2023) benchmarks. Ego-in-Exo PerceptionMCQ is derived from EgoExo4D and comprises 3,991 video question-answer (video-QA) pairs spanning four categories: action understanding (Action Und.), task-relevant region understanding (Task Regions), human-object interactions (HOI), and hand identification (Hand Ident.). Because it is derived from EgoExo4D, Ego-in-Exo PerceptionMCQ can be evaluated from either the egocentric or the exocentric viewpoint. For the ego target domain experiments, we report results using the egocentric videos, denoted as Ego-in-Exo PerceptionMCQ (Ego RGB). EgoSchema consists of 5,031 egocentric video-QA pairs derived from the Ego4D dataset (Grauman et al., 2022).

For evaluation on the **source domain**, we select benchmarks that measure exocentric video understanding capability. Specifically, we evaluate on the NeXTQA (Xiao et al., 2021), VideoMME (Fu et al., 2025), and ADL-X (Reilly et al., 2025a) benchmarks. NeXTQA and VideoMME are general-purpose video-QA benchmarks built from web-scraped videos (e.g., from YouTube), with 8,564 QA

pairs in NeXTQA and 2,700 QA pairs in VideoMME. ADL-X is a video-QA benchmark built from videos of activities of daily living, it contains a total of 10,561 multiple-choice questions (ADL-X MCQ) and 1,862 video description questions (ADL-X Desc) derived from various activities of daily living datasets (Das et al., 2019; Sigurdsson et al., 2016; Jia et al., 2020; Dai et al., 2022).

**Results.** Table 1 reports results of adaptation to the egocentric viewpoint. Training only the vision-language connector or the connector together with LLM LoRA adapters does not lead to effective adaptation to the target domain ($\Delta_{target} < 1$). Updating all three modules (connector, vision encoder, and LLM) improves performance on the target domain by $\Delta_{target} = +0.57$, but the large number of trainable parameters results in forgetting on the source benchmarks ($\Delta_{source} = -0.50$). In contrast, updating the connector and vision encoder alone slightly improves performance on the target domain and does not lead to forgetting on the source domain. E1: These results highlight that the core difficulty of domain adaptation in existing VLMs arises from the necessity of updating the vision encoder to learn domain-specific visual representations, which inevitably leads to forgetting of pretrained knowledge. Our proposed VISCOP achieves the strongest adaptation performance, with the highest improvement on the target domain ($\Delta_{target} = +3.5$) while simultaneously maintaining retention on the source benchmarks ($\Delta_{source} = +1.8$). Interestingly, VISCOP not only avoids catastrophic forgetting but also improves performance on some source benchmarks (e.g., ADL-X). We attribute this positive transfer to a multi-axis domain shift: although source and target differ in viewpoint (exocentric vs. egocentric), their action distributions overlap. ADL-X, while exocentric, encapsulates activities of daily living that closely aligns with the EgoExo4D action distribution, enabling beneficial cross-domain generalization.

Table 2: **Depth Video Understanding Experts.** Performance of adaptation strategies on the depth target domain and RGB source domain. Adaptation strategy notation follows Table 1 (✓ = trainable, ✗ = frozen). $\Delta_{target}$ and $\Delta_{source}$ denote relative gains over the Base VLM.

| Adaptation Strategy | | | Depth Benchmarks (Target) | | | | | RGB Benchmarks (Source) | | | | | | Adaptation Metrics | |
|---|---|---|---|---|---|---|---|---|---|---|---|---|---|---|---|
| | | | *Ego-in-Exo PerceptionMCQ (Exo Depth)* | | | | | Ego-in-Exo | NeXTQA | VideoMME | ADL-X | ADL-X | Avg | $\Delta_{target}$ | $\Delta_{source}$ |
| VL-C | VE | LLM | Action Und. | Task Regions | HOI | Hand Ident. | Avg | (Exo RGB) | | | MCQ | Desc | | (↑) | (↑) |
| | Base VLM | | 34.73 | 50.61 | 35.06 | 63.06 | 45.86 | 66.27 | **84.32** | **65.37** | **77.36** | 70.65 | 72.79 | - | - |
| ✓ | ✗ | ✗ | 55.67 | 66.59 | 62.46 | 64.49 | 62.30 | 71.36 | 83.15 | 62.41 | 70.90 | 69.05 | 71.37 | 16.44 | -1.42 |
| ✓ | ✓ | ✗ | **57.20** | 69.63 | 54.43 | **64.48** | 61.44 | 60.97 | 82.89 | 62.00 | 71.48 | 67.26 | 68.92 | 15.57 | -3.87 |
| ✓ | ✗ | LoRA | 42.94 | 53.54 | 43.92 | 63.96 | 51.09 | 60.97 | 83.73 | 64.19 | 72.19 | 72.49 | 70.71 | 5.23 | -2.08 |
| ✓ | VISCOP | LoRA | 56.78 | **73.17** | **66.23** | 64.35 | **65.13** | 71.89 | 83.91 | 64.30 | 76.59 | 76.47 | 74.63 | **+19.27** | **+1.84** |

### 5.2.2 DEPTH VIDEO UNDERSTANDING

**Target and source benchmarks.** For evaluation on the **target domain**, we evaluate on the Ego-in-Exo PerceptionMCQ (Reilly et al., 2025b) benchmark. In the depth-adaptation setting, we train on depth maps of exocentric videos extracted with DepthAnythingV2 (Yang et al., 2024) and evaluate on exocentric depth videos following Reilly et al. (2025b), denoted Ego-in-Exo PerceptionMCQ (Exo Depth). For the **source domain**, we use *RGB* benchmarks of exocentric understanding: Ego-in-Exo PerceptionMCQ (Exo RGB), NeXTQA, VideoMME, and ADL-X.

**Results.** We present the results for adaptation to the depth modality in Table 2. In contrast to the results on egocentric viewpoint adaptation, we find that all training strategies achieve improvements on the target domain, reflecting the disparity of the visual embedding space between the depth and RGB modalities. We find that this disparity leads to different behavior across training strategies. Jointly updating the vision encoder and the vision-language connector preserves source performance for egocentric adaptation but causes severe catastrophic forgetting under depth adaptation ($\Delta_{source} = -3.87$). This arises from the substantial encoder updates required to bridge RGB and depth, which overwrite RGB representations. In contrast, VISCOP preserves RGB features and source performance while achieving the largest target domain gains ($\Delta_{target} = +19.27$).

Table 3: **Robot Control Experts (Simulation).** Performance of adaptation strategies on the robotic control target domain and human understanding source domain. Table notation follows Table 1.

| Adaptation Strategy | | | Robotic Control Benchmarks (Target) | | | | Human Understanding Benchmarks (Source) | | | | | | Adaptation Metrics | |
|---|---|---|---|---|---|---|---|---|---|---|---|---|---|---|
| | | | *VIMA Bench* | | | | Ego-in-Exo | NeXTQA | VideoMME | ADL-X | ADL-X | Avg | $\Delta_{target}$ (↑) | $\Delta_{source}$ (↑) |
| VL-C | VE | LLM | L1 | L2 | L3 | Avg | (Exo RGB) | | | MCQ | Desc | | | |
| | Base VLM | | 0 | 0 | 0 | 0 | 66.27 | 84.32 | 65.37 | 77.36 | 70.65 | 72.79 | - | - |
| ✓ | ✓ | ✓ | 69.62 | 60.77 | 65.00 | 65.13 | 56.92 | 83.24 | 62.74 | 52.21 | 64.50 | 63.92 | +65.13 | -8.87 |
| ✓ | ✗ | ✓ | 63.46 | 63.08 | 68.75 | 65.10 | 59.42 | 83.16 | 64.41 | 52.92 | 64.86 | 64.95 | +65.10 | -7.84 |
| ✓ | VISCOP | ✓ | 67.69 | **65.77** | 70.00 | **67.82** | 71.19 | 83.71 | 63.67 | 55.89 | 66.62 | 68.22 | +67.82 | -4.58 |

### 5.2.3 ROBOT CONTROL

**Target and source benchmarks.** For evaluation on the **target domain**, we consider both simulated and real-world robotic environments. In simulation, we use the evaluation set of VIMA-Bench (Jiang et al., 2023), which organizes tasks into three levels of difficulty: L1 (Object Placement), where all objects have been seen during training; L2 (Novel Combination), where objects seen during training appear in new pairings or contexts; and L3 (Novel Objects), where objects entirely unseen during training are introduced. Together, these levels measure generalization from familiar training conditions to progressively more challenging distributions. In the real-world setting, we evaluate on three tabletop manipulation tasks: T1) Place the {object} on the plate, T2) Pick up and rotate {object} by {angle}; and T3) Move all {color} objects onto the plate. Examples of each task and a list of objects used is provided in Appendix A.2. E4: On these robotic control benchmarks, the reported accuracy corresponds to the success rate across all robot manipulation tasks. For **source domain** evaluation of VLMs trained on both real and simulated robotic environments, we use the human-activity video benchmarks Ego-in-Exo (Exo RGB), NeXTQA, VideoMME, and ADL-X.

**Results.** The results of adaptation to the robotic control domain are presented in Table 3. E5: The base VLM demonstrates weak performance on all robot control tasks, as its pretraining distribution lacks action trajectories (i.e., instruction data mapping from visual observations to robot actions). This lack of pretraining results in 0% accuracy across all levels of VIMA-Bench, and is consistent with prior works (Li et al., 2025). This highlights the ex-

Table 4: **Robot Control Experts (Real-world)** Performance on the robotic control target domain and human understanding source domain.

| Adaptation Strategy | | | Robotic Control Benchmarks (Target) | | | | Adaptation Metrics | |
|---|---|---|---|---|---|---|---|---|
| VL-C | VE | LLM | T1 | T2 | T3 | Avg | $\Delta_{target}$ (↑) | $\Delta_{source}$ (↑) |
| *Training data: VIMA-Bench* | | | | | | | | |
| ✓ | ✓ | ✓ | **45.00** | 60.00 | 15.00 | 40.00 | +40.00 | -8.87 |
| ✓ | VISCOP | ✓ | 40.00 | **70.00** | 20.00 | **43.33** | **+43.33** | **-4.58** |
| *Training data: VIMA-Bench + xArm-Det* | | | | | | | | |
| ✓ | ✓ | ✓ | 85.00 | 85.00 | 70.00 | 80.00 | +80.00 | -11.04 |
| ✓ | VISCOP | ✓ | **100.00** | **100.00** | **90.00** | **96.67** | **+96.67** | **-11.00** |

treme domain gap both in the visual space (robot observations vs. human videos) and in the language space (control actions vs. linguistic outputs) between the source and target domains. Similarly to the depth adaptation setting, we find that training the vision encoder improves performance on the target domain, but results in the worst source domain retention ($\Delta_{source} = -8.87$) of all robot control experts. In contrast, our proposed VISCOP achieves the best performance on the target domain ($\Delta_{target} = +67.82$) while retaining the most source domain knowledge ($\Delta_{source} = -4.58$) compared to other experts, demonstrating the effectiveness of our method even when the gap between the source and target domains is very large. Also note that VISCOP operates on per-timestep images in these experiments; thus the visual probes consume the same visual tokens as the vision encoder, suggesting they extract domain-specific representations more effectively than the base vision encoder.

We further evaluate adaptation in the real-world setting using the xArm-Det dataset in Table 4. We consider a *transfer setting*, where the experts are trained only on VIMA-Bench and directly evaluated on xArm-Det, and the setting where the experts are jointly trained on both VIMA-Bench and xArm-Det. In both cases, our proposed VISCOP outperforms the vision encoder trained experts on target domain adaptation as well as source domain retention.

### 5.3 MODEL DIAGNOSIS AND ANALYSIS

In this section, we motivate the design of VISCOP through a diagnostic study, and perform an analysis on the visual representations it learns. We investigate the number of visual probes, as well as the placement of interaction modules within the vision encoder. We then analyze the domain-specific representations learned by VISCOP through t-SNE and attention visualizations.

**Alternatives to learnable queries.** Table 5 compares VISCOP against alternative adaptation strategies. *Visual Probes Only (VP)* trains only visual probes with their vision-language connector ($\mathcal{C}_{probe}$) without any interaction modules. *Partial Encoder Training (Last-4)* makes the final four layers of the vision encoder trainable. *QFormer-Style Compression* uses visual probes with interaction modules only at the vision encoder's final layer, mimicking Q-Former's compression approach (Li et al., 2023b). E9: *Model Tailor* (Zhu et al., 2024) performs post-hoc domain adaptation by fusing parameter updates from a fine-tuned VLM back into the base VLM, modifying only the LLM parameters and leaving the vision encoder untouched. Training with QFormer-Style compression or only training with visual probes (*VP*) underperforms compared to VISCOP, indicating the importance of probe interactions at intermediate layers of the vision encoder to learn domain-specific features across multiple levels of abstraction. Similarly, training only the last four layers of the vision encoder, or training it with LoRA, also underperforms, highlighting that even partial parameter updates fail to capture domain-specific signals as effectively as VISCOP's visual probes.

Table 5: **Ablation on alternative adaptation approaches.** Annotation legend: *VP* (visual probes and probe connector with no interaction modules), *Last-4* (train only the last 4 vision encoder layers), *QFormer Style.* (interaction module is placed only at the last layer of the VE). *Model Tailor* (adaptation approach proposed in Zhu et al. (2024)).

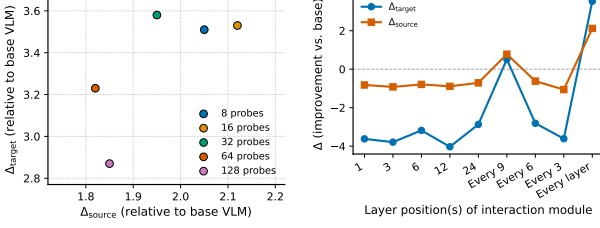

| Adaptation Strategy | | | Target | Source | Adaptation Metrics | |
|---|---|---|---|---|---|---|
| VL-C | VE | LLM | Avg | Avg | $\Delta_{target}$ (↑) | $\Delta_{source}$ (↑) |
| | Base VLM | | 70.43 | 74.42 | – | – |
| ✓ | *VP* | LoRA | 65.57 | 75.05 | -4.86 | +0.62 |
| ✓ | LoRA | LoRA | 69.85 | 75.35 | -0.59 | +0.92 |
| ✓ | *Last-4* | LoRA | 70.46 | 72.62 | +0.02 | -1.80 |
| ✓ | *QFormer Style* | LoRA | 70.99 | 75.03 | +0.56 | +0.61 |
| ✓ | ✗ | Model Tailor | 70.27 | 75.29 | -0.16 | +0.86 |
| ✓ | VISCoP | LoRA | **73.96** | **75.74** | **+3.53** | **+2.12** |

Figure 3: **Ablation on the number of visual probes in VISCoP.** We explore 8, 16, 32, 64, and 128 probes on the egocentric domain.

Figure 4: **Ablation on positions of interaction modules in VISCoP.** Results are presented on the egocentric target domain.

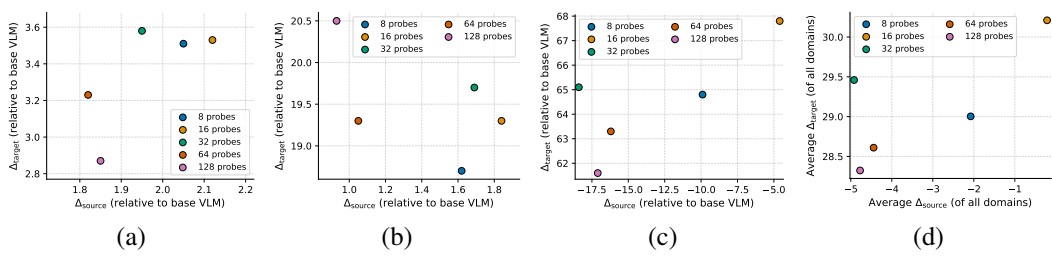

(a)      (b)      (c)      (d)

Figure 5: **Effect of number of visual probes across domains.** (a) Egocentric viewpoint, (b) Depth modality, (c) Robotic control, (d) Average over all three target domains.

**E9:** Model Tailor also falls short in this setting, suggesting that approaches which do not leverage intermediate vision encoder representations struggle to learn domain-specific visual features.

**E2: Computation overhead of VISCoP.** VISCoP introduces only modest computational overhead relative to the base VLM, introducing only 2% more parameters than the base VLM. In Table 6, we compute

Table 6: **Computation overhead of VisCoP.**

| Model | Max VRAM | Inference Latency (Entire Model) | Inference Latency (Before LLM) | Total Num Params. |
|---|---|---|---|---|
| Base VLM | 24.4GB | 0.767s | 0.056s | 8.04B |
| VISCoP | 27.8GB | 0.417s | 0.069s | 8.21B |

the average VRAM usage and latency during inference on the Ego-in-Exo PerceptionMCQ benchmark, as well as the total parameter count. We find that VISCoP increases VRAM usage by 3.4GB and adds just 0.013s of inference latency in visual feature extraction (Before LLM). Interestingly, the inference latency of our model is lower than that of the base VLM. We attribute this to the base VLM producing longer, less focused responses, which increases total decoding time.

**Ablations on probes and interaction modules.** We study the effect of the number of visual probes and the placement of interaction modules (Figure 3, Figure 4). Probes consistently improve performance over the base VLM, with the best trade-off at 16 probes ($\Delta_{target} = +3.53$, $\Delta_{source} = +2.12$); larger probe counts offer no further gains and can reduce performance due to redundancy. For interaction modules, applying them at every encoder layer yields the strongest adaptation, while sparse placement (e.g., every 6 or 9 layers) provides weaker or inconsistent gains. These results highlight the importance of using a small number of probes with dense access to intermediate features.

**E8:** In Figure 5, we examine the effect of varying the number of visual probes across all three target domains. We verify that 16 probes provides a fair overall tradeoff across all target domains, retaining the most source-domain performance across all domains while remaining near the top in egocentric and depth adaptation, and performing substantially better than larger probe counts in the robotics setting. We attribute this to the fact that robotics tasks require the model to integrate visual cues with precise action semantics—when the number of probes becomes large, the additional probe signals tend to dominate the representation space, causing the LLM to overcommit to action-execution patterns and generate robotic command-like outputs even when inappropriate. In contrast, a smaller probe set provides focused domain-specific visual information without overwhelming the pretrained multimodal alignment, resulting in stronger performance and better retention.

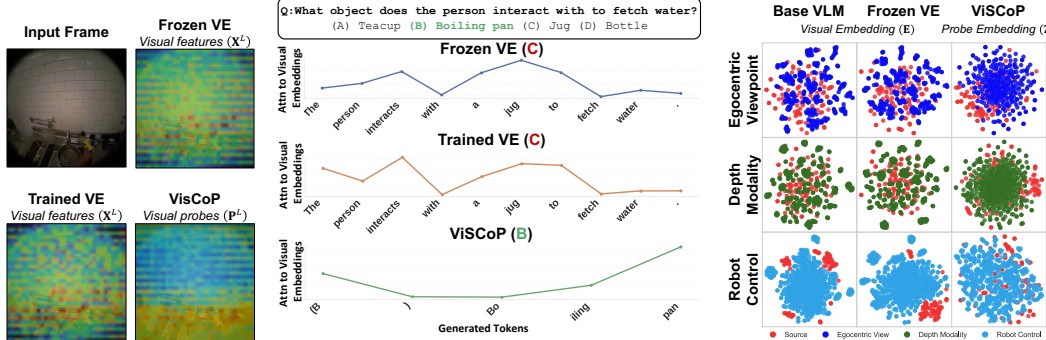

(a) **Attention visual of visual features and probes.**  (b) **Attentions of generated language tokens to visual embeddings.**  (c) **t-SNE visualization of source and target domain features.**

Figure 6: **Analysis of the learned representations of VISCOP.** (a) Visualization of attentions between visual features and visual probes. (b) Attention of generated language tokens to visual embeddings. (c) t-SNE projection of source and target domain features.

**Visualizing attention in domain-adapted VLMs**  In Figure 6a, we analyze attention maps of various VLM adaptation strategies to assess how different components capture domain-specific visual features. For both the frozen and trainable vision encoders, we visualize attention using attention rollout (Abnar & Zuidema, 2020), for VISCOP we visualize the attentions of the visual probes, averaged across all probes. The frozen vision encoder fails to focus consistently on relevant regions under the experimented domains, reflecting its limited ability to capture domain-specific features. The trained vision encoder yields sharper attention on the relevant regions, indicating its ability to learn domain-specific features, albeit at the cost of catastrophic forgetting of the source domain as shown in Section 5.2. In contrast, the visual probes of VISCOP have a sharp focus on the task-relevant regions, despite the vision encoder being frozen. This indicates that the probes alone are able to extract the domain-specific visual features necessary for adaptation. In Figure 6b, we visualize the attention of generated language tokens to visual embeddings. We find that VISCOP correctly responds to the query, with more focus given to tokens corresponding to relevant objects.

**Learning domain-specific representations.**  Figure 6c compares t-SNE embeddings from three models: the base VLM, a VLM adapted with a frozen vision encoder, and VISCOP. In the egocentric and depth domains, both the base and frozen-encoder VLMs entangle the embeddings of source and target domains, indicating that they fail to capture domain-specific structure. In contrast, the visual probes of VISCOP are able to learn domain-relevant features without requiring updates to the vision encoder, producing well-separated clusters for the source and target domains, despite using only 16 probes. For robotics, however, the trend reverses: the base and frozen models form distinct robot clusters, whereas VISCOP learns a more compact, entangled representation.

## 6 CONCLUSION

We introduced VISCOP, a mechanism that extracts domain-specific visual features through probing of a frozen vision encoder to enable effective domain adaptation in VLMs and prevent catastrophic forgetting. VLMs equipped with VISCOP achieve superior target domain performance, while maintaining strong source domain capabilities across cross-view, cross-modal, and cross-task adaptation scenarios. We will release all code, models, and evaluation protocols to facilitate future research.

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

## A APPENDIX

### A.1 DETAILS OF SIMULATED ROBOT CONTROL EXPERIMENTS

For our robot control simulation experiments, we use the VIMA-8K instruction set generated from the VIMA dataset, following (Li et al., 2025). Figure 7 illustrates representative examples of training tasks - simple visual manipulation (top row) and rotation (middle row).

For evaluation, we adopt the three levels of generalization defined in VIMA-Bench (Jiang et al., 2023): - **L1 (Placement Generalization):** tasks where the object placements differ from those seen in the training set. - **L2 (Combination Generalization):** tasks requiring new combinations of objects not paired during training. - **L3 (Novel Object Generalization):** tasks involving completely unseen objects that were not present in the training data.

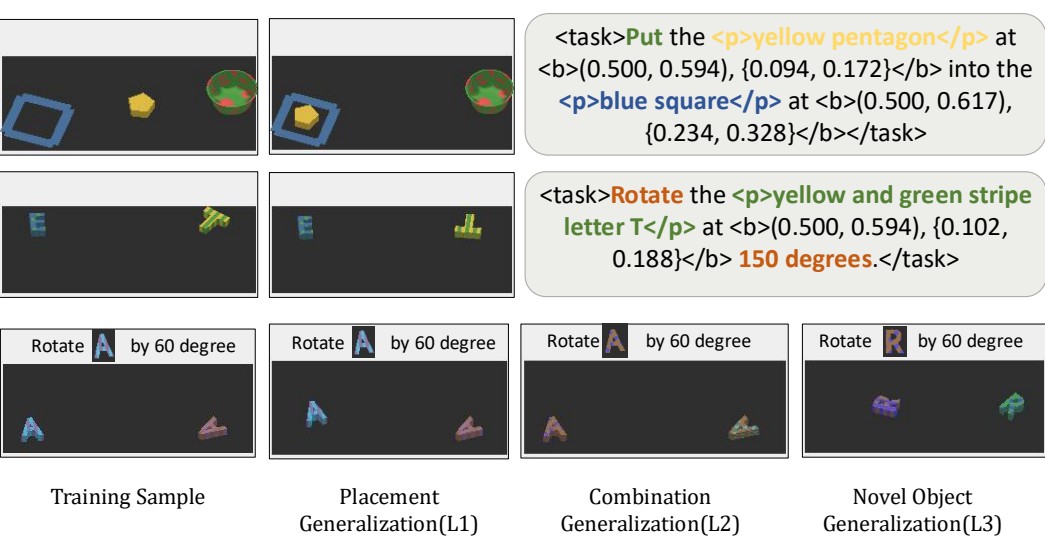

Training Sample      Placement Generalization(L1)      Combination Generalization(L2)      Novel Object Generalization(L3)

Figure 7: **Examples from VIMA and VIMA-Bench**. The first two rows show training examples, including the initial observations, final states, and task instructions. The bottom row illustrates the evaluation in VIMA-Bench, covering three levels of generalization.

### A.2 DETAILS OF REAL-WORLD ROBOTICS EXPERIMENTS

We provide additional details of the experiments conducted in our novel robot environment, including the setup, data collection, and evaluation protocol.

#### A.2.1 REAL-ROBOT SETUP

Our setup consists of an xArm7 robotic arm with a gripper, tabletop, and an Intel RealSense D455 third person camera mounted in front of the arm to collect observations as seen in Figure 8. The action space of the end effector is two 2D cartesian coordinates representing the pick and place poses, and two quaternions for rotations similar to (Jiang et al., 2023). We evaluated the effectiveness of our method mainly on three robot manipulative tasks:

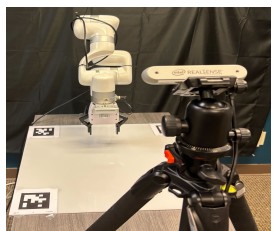

Figure 8: **Real robot setup.** Our setup uses an xArm7 robot arm and Intel RealSense D455 camera.

**T1** : Place the {object} on the plate. **T2** : Pickup and Rotate the {object} by {degree} degrees. **T3** : Move all the {colour} objects into the plate.

We uniformly sample {object} from a set of 10 toys : green apple, carrot, eggplant, banana, corn, grape, green pepper, tomato, strawberry, cucumber, clementine, and lemon. For T2, the target

rotation angle is randomly selected from {30°, 45°, 60°, 90°, 180°}. For T3, the variable `colour` is chosen from four categories: {red, orange, yellow, purple}

### A.2.2 REAL-ROBOT DATA COLLECTION

We collected 1,007 images with resolution 640 x 640 of a real-robot setup with multiple objects scattered on the table. A one-shot object detection using Owlv2 (Matthias Minderer, 2023) is applied to extract bounding boxes for each object. Based on these images and their corresponding bounding box annotations, we generate task instructions following the xArm-Det style similar to (Li et al., 2025)

### A.2.3 EVALUATION PROTOCOL

All three tasks are evaluated under two settings: zero-shot and joint training. The observation space is illustrated in Figure 9. In Zero-shot setting, we use the models trained on Vima data where as in the joint training setting, we tune VLM jointly on both Vima data and collected xArm-Det data. For each task, we conduct 20 trials with objects placed at random initial positions on the table. Each episode is limited to a maximum of 4 steps. We report the average success rate across all trials as performance metric and below are the success criteria for each task that we follow :

T1 : A trial is considered successful if at least 50% of the object lies inside the plate.

T2 : A trail is successful by visually verifying whether the object has been rotated to the specified target angle.

T3 : A trial is successful only if all objects of the specified color are placed into the plate; otherwise, it is a failure.

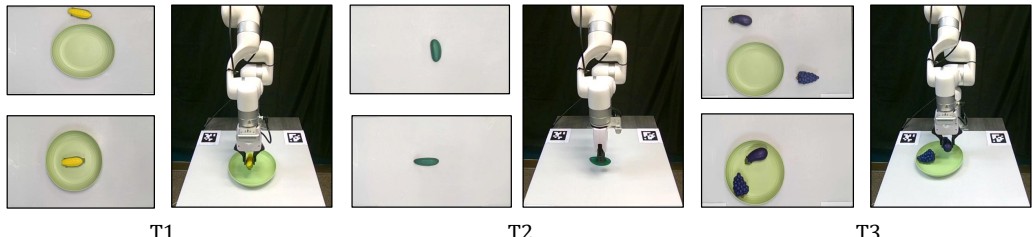

Figure 9: **Three real-world robot tasks.** Each column shows the initial state (top) and the corresponding final state (bottom), along with the robot execution (from left to right): **T1** (place the corn on the plate), **T2** (rotate the cucumber by 90°), and **T3** (move all purple objects into the plate).

## B  EXPANDED EXPERIMENTAL RESULTS

In this section, we present additional experimental results, and expanded results on the ADL-X benchmark across three target domains: **ego-video understanding** Table 7, **depth-video understanding** Table 8, and **robot control** Table 9. In addition, we provide comprehensive source-domain results for the real-world domain expert Table 10, as well as detailed ablation studies Table 11. For the ADL-X description benchmark, we restrict evaluation to the Charades Description (Reilly et al., 2025a).

Table 7: **Performance of ego video expert on ADL-X Benchmark.**

| Adaptation Strategy | | | ADL-X MCQ | | | | | ADL-X Descriptions (Charades) | | | | | |
|---|---|---|---|---|---|---|---|---|---|---|---|---|---|
| VL-C | VE | LLM | Charades AR | Smarthome AR | TSU TC | LEMMA TC | Avg | Cor | Do | Ctu | Tu | Con | Avg |
| | Base VLM | | 91.95 | 70.58 | 78.34 | 68.56 | 77.36 | 73.50 | 73.74 | 75.78 | 68.59 | 61.61 | 70.64 |
| ✓ | ✗ | ✗ | 93.10 | 70.34 | 75.73 | 67.04 | 76.55 | 79.30 | 80.82 | 82.43 | 73.13 | 61.82 | 75.50 |
| ✓ | ✓ | ✗ | 91.56 | 71.48 | 77.16 | 67.99 | 77.05 | 80.55 | 81.55 | 83.57 | 73.56 | 61.20 | 76.09 |
| ✓ | ✗ | LoRA | 92.39 | 71.50 | 77.59 | 68.18 | 77.41 | 78.54 | 77.34 | 81.70 | 73.45 | 60.74 | 74.36 |
| ✓ | VISCOP | LoRA | 92.83 | 72.26 | 82.60 | 68.18 | 78.97 | 79.82 | 82.65 | 83.86 | 74.70 | 62.82 | 76.77 |

Table 8: **Performance of depth video expert on ADL-X Benchmark.**

| Adaptation Strategy | | | ADL-X MCQ | | | | | ADL-X Descriptions (Charades) | | | | | |
|---|---|---|---|---|---|---|---|---|---|---|---|---|---|
| VL-C | VE | LLM | Charades AR | Smarthome AR | TSU TC | LEMMA TC | Avg | Cor | Do | Ctu | Tu | Con | Avg |
| | Base VLM | | 91.95 | 70.58 | 78.34 | 68.56 | 77.36 | 73.50 | 73.74 | 75.78 | 68.59 | 61.61 | 70.64 |
| ✓ | ✗ | ✗ | 90.84 | 56.26 | 71.51 | 64.96 | 70.89 | 71.22 | 75.31 | 75.95 | 65.80 | 56.96 | 69.05 |
| ✓ | ✓ | ✗ | 90.90 | 54.55 | 73.65 | 66.47 | 71.47 | 69.96 | 73.60 | 73.83 | 64.04 | 54.82 | 67.25 |
| ✓ | ✗ | LoRA | 91.34 | 57.55 | 73.94 | 65.90 | 72.18 | 77.50 | 77.40 | 79.58 | 69.35 | 58.58 | 72.48 |
| ✓ | VISCOP | LoRA | 93.60 | 63.79 | 81.71 | 67.23 | 76.58 | 78.51 | 84.68 | 84.07 | 74.67 | 60.41 | 76.47 |

Table 9: **Performance of robot control expert on ADL-X Benchmark.**

| Adaptation Strategy | | | ADL-X MCQ | | | | | ADL-X Descriptions (Charades) | | | | | |
|---|---|---|---|---|---|---|---|---|---|---|---|---|---|
| VL-C | VE | LLM | Charades AR | Smarthome AR | TSU TC | LEMMA TC | Avg | Cor | Do | Ctu | Tu | Con | Avg |
| | Base VLM | | 91.95 | 70.58 | 78.34 | 68.56 | 77.36 | 73.50 | 73.74 | 75.78 | 68.59 | 61.61 | 70.64 |
| ✓ | ✓ | ✓ | 78.05 | 36.24 | 36.39 | 58.14 | 52.21 | 66.16 | 68.30 | 70.66 | 61.78 | 55.6 | 64.50 |
| ✓ | ✗ | ✓ | 78.88 | 39.81 | 35.61 | 57.38 | 52.95 | 66.54 | 68.95 | 71.03 | 62.58 | 55.15 | 64.85 |
| ✓ | VISCOP | ✓ | 90.96 | 45.77 | 38.76 | 48.1 | 55.89 | 66.25 | 71.91 | 72.49 | 66.02 | 56.433 | 66.62 |

Table 10: **Expanded Robot Control Experts (Real-world)**

| Adaptation Strategy | | | Robotic Control Benchmarks | | | | Human Understanding Benchmarks | | | | | | Adaptation Metrics | |
|---|---|---|---|---|---|---|---|---|---|---|---|---|---|---|
| VL-C | VE | LLM | T1 | T2 | T3 | Avg | Ego-in-Exo (Exo RGB) | NeXTQA | VideoMME | ADL-X MCQ | ADL-X Desc | Avg | Δ_target (↑) | Δ_source (↑) |
| | Base VLM | | 0 | 0 | 0 | 0 | 66.27 | 84.32 | 65.37 | 77.36 | 70.65 | 72.79 | - | - |
| *Training data: VIMA-Bench* | | | | | | | | | | | | | | |
| ✓ | ✓ | ✓ | 45.00 | 60.00 | 15.00 | 40.00 | 56.92 | 83.24 | 62.74 | 52.21 | 64.50 | 63.92 | +40.00 | -8.87 |
| ✓ | VISCOP | ✓ | 40.00 | 70.00 | 20.00 | 43.33 | 71.19 | 83.71 | 63.67 | 55.89 | 66.62 | 68.22 | +43.33 | -4.58 |
| *Training data: VIMA-Bench + xArm-Det* | | | | | | | | | | | | | | |
| ✓ | ✓ | ✓ | 85.00 | 85.00 | 70.00 | 80.00 | 64.50 | 83.00 | 63.00 | 36.04 | 62.24 | 61.76 | +80.00 | -11.04 |
| ✓ | VISCOP | ✓ | 100.00 | 100.00 | 90.00 | 96.67 | 59.59 | 82.98 | 63.26 | 36.32 | 66.83 | 61.79 | +96.67 | -11.00 |

Table 11: **Comprehensive target-source domain results from the ablation study of VISCOP**

| Adaptation Strategy | | | Egocentric Benchmarks | | | | | | Exocentric Benchmarks | | | | | Adaptation Metrics | |
|---|---|---|---|---|---|---|---|---|---|---|---|---|---|---|---|
| | | | *Ego-in-Exo PerceptionMCQ (Ego RGB)* | | | | | | | | | | | | |
| VL-C | VE | LLM | Action Und. | Task Regions | HOI | Hand Ident. | EgoSchema | Avg | NeXTQA | VideoMME | ADL-X MCQ | ADL-X Desc | Avg | Δ_target (↑) | Δ_source (↑) |
| | Base VLM | | 75.37 | 74.88 | 75.56 | 65.38 | 60.98 | 70.43 | 84.32 | 65.37 | 77.36 | 70.65 | 74.42 | - | - |
| ✓ | VP | LoRA | 66.88 | 75.98 | 59.62 | 63.84 | 61.54 | 65.57 | 84.22 | 64.37 | 77.86 | 73.73 | 75.05 | -4.86 | 0.62 |
| ✓ | LoRA | LoRA | 73.76 | 75.24 | 73.55 | 64.99 | 61.68 | 69.85 | 84.22 | 64.48 | 77.52 | 75.17 | 75.35 | -0.59 | 0.92 |
| ✓ | last-4 | LoRA | 73.35 | 77.93 | 73.32 | 65.25 | 62.43 | 70.46 | 84.00 | 63.78 | 77.74 | 76.34 | 72.62 | 0.02 | -1.80 |
| ✓ | QFormer-Style | LoRA | 75.99 | 77.56 | 74.50 | 65.38 | 61.54 | 70.99 | 84.13 | 64.44 | 78.43 | 73.13 | 75.03 | 0.56 | 0.61 |
| ✓ | VISCOP | LoRA | 81.28 | 82.80 | 78.75 | 64.86 | 62.11 | 73.96 | 84.31 | 64.70 | 78.97 | 76.78 | 76.19 | +3.53 | +1.77 |

## C  QUALITATIVE RESULTS

In this section, we provide qualitative comparisons of three models—Base VLM, trained vision encoder (VL-C+VE), and VISCOP across the three domain experts ego-video understanding, depth-video understanding, and robot control. Figures 10, 11, and 12 show representative examples from each expert. Each figure shows representative samples from both the target domain and the source domain.

We demonstrate that VL-C+VE successfully adapts the Base VLM to the target domain, enabling correct predictions. However, this adaptation comes at the expense of source-domain performance, where VL-C+VE frequently makes mistakes. In contrast, VISCOP achieves the best of both: it adapts effectively to the target domain while simultaneously retaining strong performance on the source domain, thereby avoiding catastrophic forgetting.

We also provide qualitative comparisons of video descriptions on the source domain (ADL-X) using the ego-video understanding expert and the depth-video understanding expert. As shown in Figure 13 and Figure 14, our method generates descriptions that are both more accurate and more detail-oriented compared to the trained vision encoder (VL-C+VE). While VL-C+VE can adapt to the target domain, on the source domain it often introduces hallucinated details. In contrast, VISCOP preserves correctness, capturing the scene, actions and object interactions.

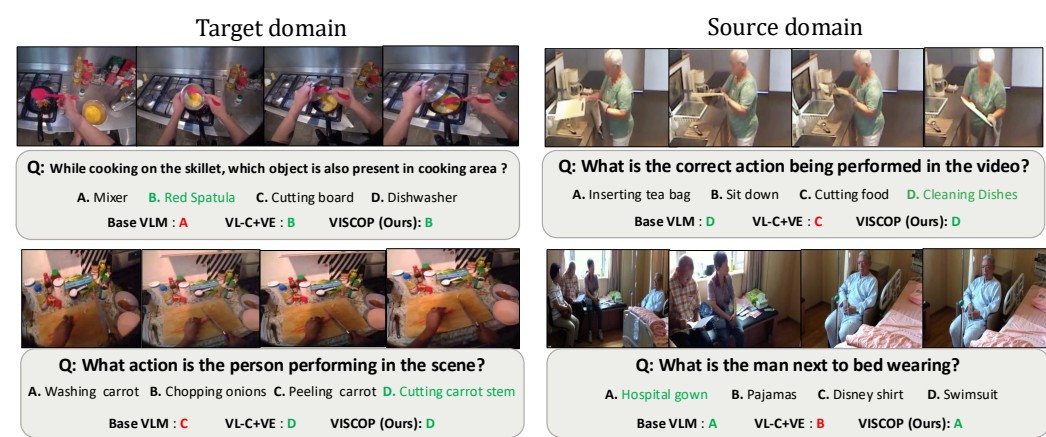

Figure 10: **Qualitative results on Egocentric Video Understanding Experts.**

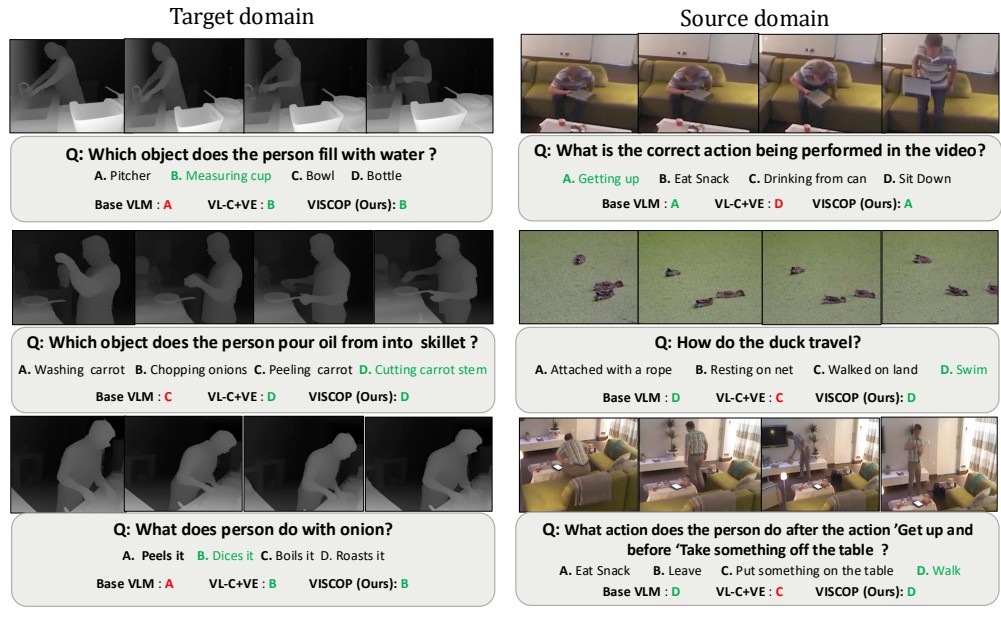

Figure 11: **Qualitative results on Depth Video Understanding Experts.**

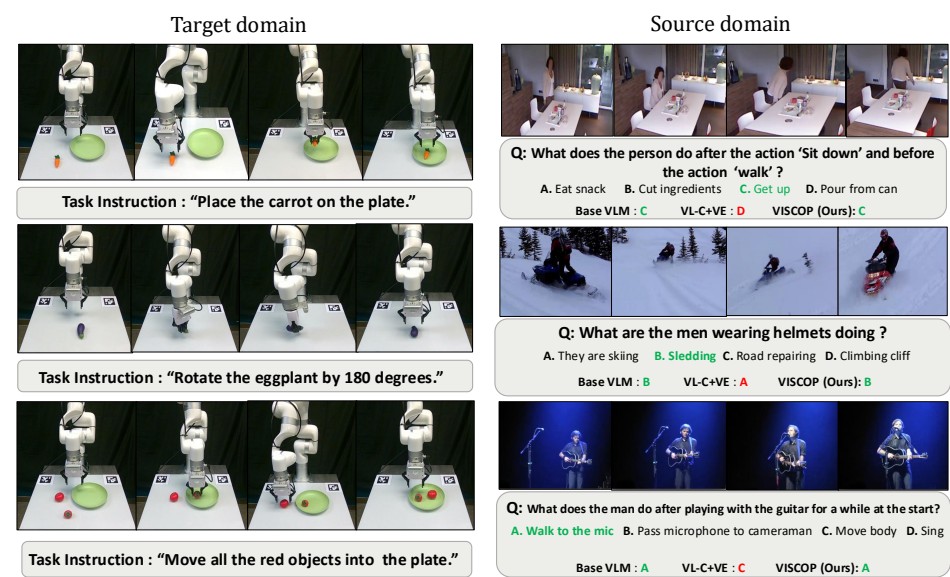

Figure 12: **Qualitative results on robot control Experts.**

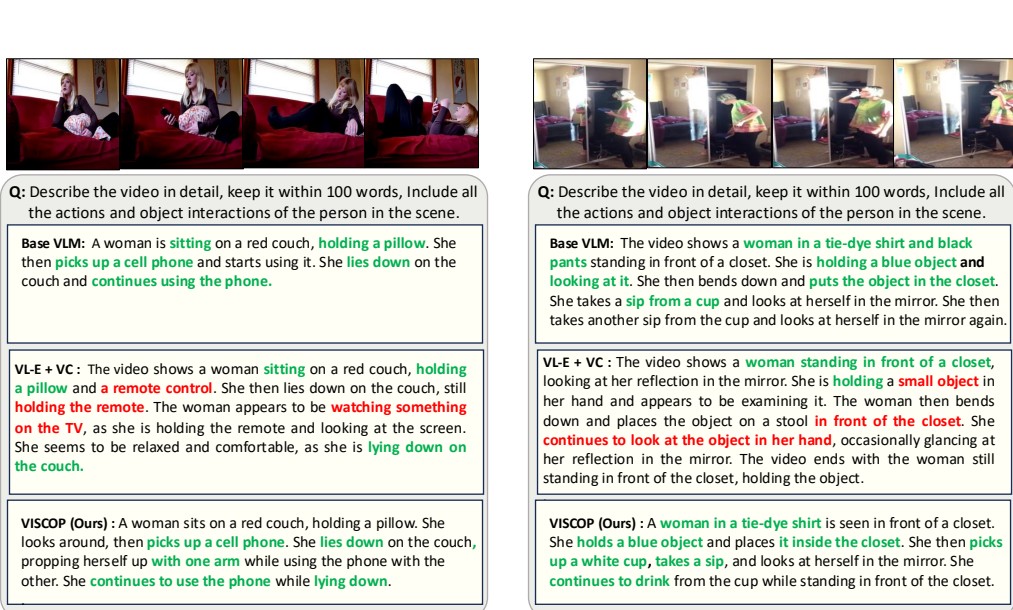

Figure 13: **ADL-X descriptions using Ego Video Understanding Expert.**

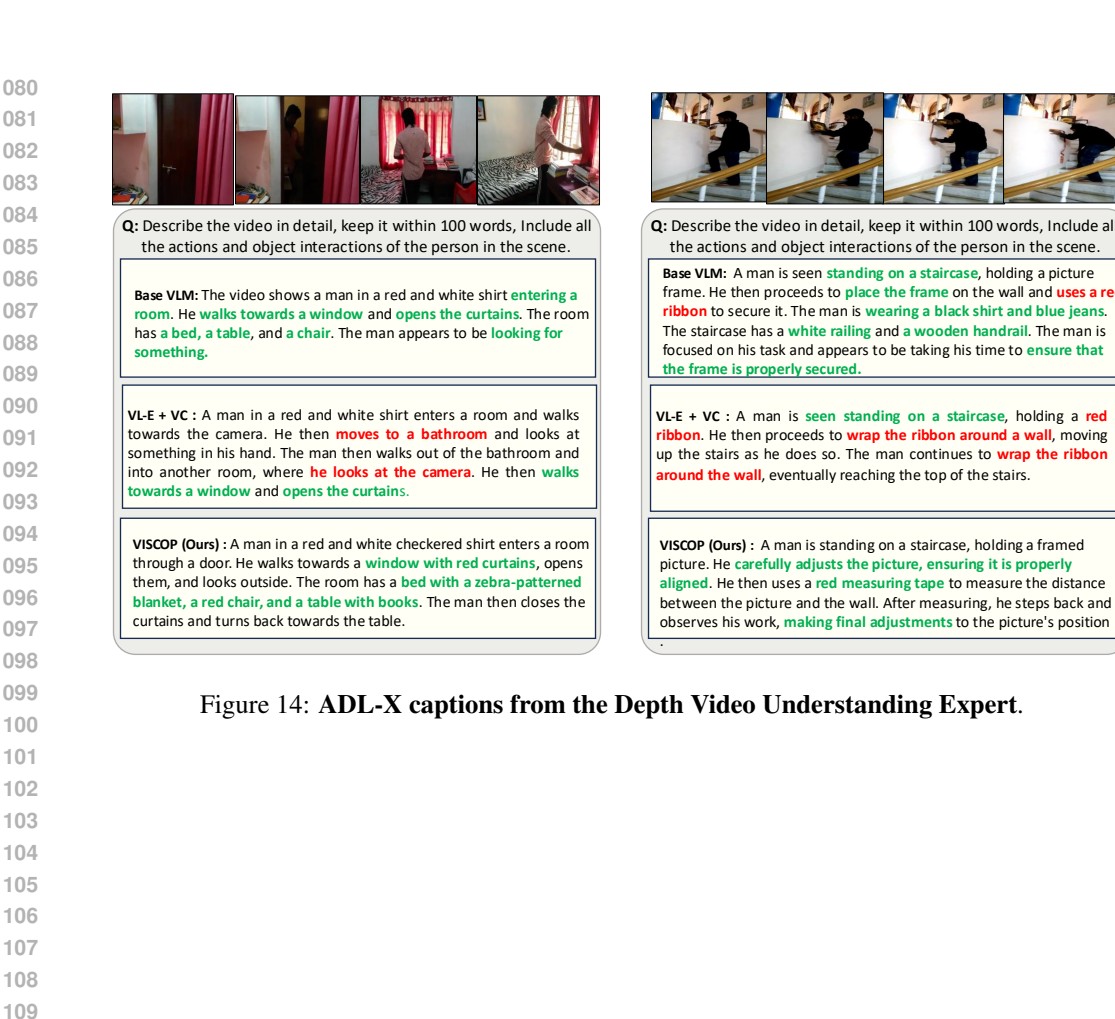

Figure 14: **ADL-X captions from the Depth Video Understanding Expert.**

