# OpenReview forum: "VisCoP: Visual Probing for Video Domain Adaptation of Vision Language Models"
_ICLR.cc/2026/Conference — Submitted to ICLR 2026_

### Official Review · Reviewer_PRAm · 2025-10-31

**Soundness:** 2
**Presentation:** 2
**Contribution:** 3
**Rating:** 6
**Confidence:** 3

**Summary:**

The paper introduces VISCOP (Vision Contextualized Probing), a method to adapt vision-language models (VLMs) to new domains—like egocentric video, depth imagery, or robotic control—without fine-tuning the frozen vision encoder. Instead, VISCOP uses a small set of learnable "visual probes" that interact with intermediate features of the frozen encoder to extract domain-specific visual cues, allowing the model to retain its original capabilities while excelling in the new domain. Experiments across three challenging adaptation tasks show that VISCOP outperforms traditional fine-tuning approaches by achieving higher target-domain accuracy while avoiding catastrophic forgetting of source-domain knowledge.

**Strengths:**

- VISCOP introduces a novel, parameter-efficient mechanism for domain adaptation in VLMs by using learnable visual probes that interact with intermediate visual hiddens, avoiding catastrophic forgetting while enabling strong target-domain performance.
- The method achieves state-of-the-art results across three challenging domain adaptation scenarios—cross-view (exocentric to egocentric), cross-modal (RGB to depth), and cross-task (human action to robotic control).
- Empirical analyses, i.e., attention visualizations, t-SNE embeddings, and ablations on probe count and layer interaction, demonstrate that VISCOP captures domain-specific visual features effectively.

**Weaknesses:**

- While VISCOP avoids finetuning the vision encoder, the method’s dependency on a fixed, frozen encoder limits its ability to exploit domain-specific low-level features that might be better learned through encoder updates.
- The paper lacks ablation on probe initialization methods and alternative interaction module architectures beyond cross-attention.
- No analysis is provided on inference latency or memory overhead introduced by the visual probes.
- Real-world robotic evaluation uses a small, non-public dataset (xArm-Det) with limited documentation on object diversity, pose variation, and camera calibration, reducing reproducibility

**Questions:**

See Weaknesses

---

> ### Author Response · Authors · 2025-11-22
> **Response to Reviewer PRAm**
>
> We sincerely thank the reviewer for recognizing the critical novelty of our visual probing approach and for their positive assessment of VisCoP's effectiveness across the three challenging domain adaptation scenarios. We appreciate the positive feedback on the comprehensiveness of both our target domain coverage and our empirical analyses demonstrating VisCoP's effectiveness at learning domain-specific visual features. We have carefully considered all points and provide detailed responses to the questions and concerns raised by the reviewer.
>
> ---
>
> > **While VISCOP avoids fine tuning the vision encoder, the method’s dependency on a fixed, frozen encoder limits its ability to exploit domain-specific low-level features that might be better learned through encoder updates.**
>
> Thank you for your insightful comment. In fact, VisCoP is motivated by this very issue, and is designed explicitly to extract these domain-specific low-level features that the frozen vision encoder can not capture (lines 69-80).
>
> ---
>
> >**The paper lacks ablation on probe initialization methods.**
>
> Thank you for your concern. **To address the concern regarding probe initialization, we perform an additional experiment exploring two alternative initialization strategies for the visual probes:** zero-initialization (zero-init) and avg vision encoder feature (Avg VE) initialization. In zero-init, we initialize all of the visual probes with zeros. In Avg VE, we initialize the probes with the average pooled visual feature across all training samples. We present the results below, comparing with our default strategy of initialization from a normal distribution (normal init) on the egocentric target domain. We find that normal initialization provides the best balance, yielding strong target-domain gains while retaining more source-domain performance, suggesting that learning probe representations from scratch is more effective than anchoring them to fixed or averaged encoder features.
>
> | Probe Init Strategy      | $\Delta_{target}$ | $\Delta_{source}$ |
> |--------------------------|-------------------|-------------------|
> | Zero init | +3.39  | +0.79            |
> | Avg VE init | +1.58             | +1.25             |
> | Normal init (default)    | +3.53             | +1.77          |
>
> ---
>
> >**The paper lacks ablation on alternative interaction module architectures beyond cross-attention.**
>
> **Regarding alternative interaction module architectures, we perform an experiment leveraging the self-attention of the vision encoder to process the visual probes as an alternative to the interaction module with cross-attention.** Our findings suggest that an interaction module leveraging self-attention does not perform as effectively as our default cross-attention variant. To address the reviewer’s concern, we update the paper and present these results in Table 5, along with a corresponding discussion in Section 5.3 (blue font, E7).
>
> ---
>
> >**No analysis is provided on inference latency or memory overhead introduced by the visual probes.**
>
> Thank you for your concern. **To address this, we have added Table 6 and updated Section 5.3 of the paper to quantify and discuss the computation overhead introduced by our proposed method (blue font, E2)**. In summary, we find that on average VisCoP requires only 3.4GB more VRAM the base VLM at inference, and only increases visual feature inference latency by 0.013s.
>
> ---
>
> >**Real-world robotic evaluation uses a small, non-public dataset (xArm-Det) with limited documentation on object diversity, pose variation, and camera calibration, reducing reproducibility**
>
> Thank you for this insightful concern. We are happy to release the xArm-Det dataset as well as corresponding camera calibration parameters; however, we doubt how useful it will be for others in the community. The xArm-Det dataset is collected specifically for our robot embodiment, meaning the dataset will only be useful to others if they are able to perfectly replicate our setup (robot arm, background, etc.). Due to the difficulty of perfect replication, it is standard practice in robotics to not release these embodiment specific datasets [1, 2, 3]. In fact, many works such as SimplerEnv [4] aim to replicate popular real-world robot settings in simulation to mitigate this issue. **To address this concern, we dedicate one page of the supplementary (Appendix A.2) to detailing the collection of xArm-Det to enable reproducibility for alternative robot embodiments.** If the reviewer believes public release of xArm-Det would benefit the community, we are fully prepared to accommodate their request.
>
> ---
>
> [1] Kim et al., "OpenVLA: An Open-Source Vision-Language-Action Model", PMLR 2025
>
> [2] Li et al., “LLaRA: Supercharging Robot Learning Data for Vision-Language Policy”, ICLR 2025
>
> [3] Black et al., "$\pi_0$: A Vision-Language-Action Flow Model for General Robot Control", arxiv 2025
>
> [4] Li et al., "Evaluating Real-World Robot Manipulation Policies in Simulation", arxiv 2024

---

### Official Review · Reviewer_z5Ck · 2025-10-31

**Soundness:** 3
**Presentation:** 4
**Contribution:** 4
**Rating:** 6
**Confidence:** 4

**Summary:**

VISCOP introduces cross attention based probing approach for domain adaptation of video understanding in VLMs. VISCOP augments a frozen vision encoder with a compact set of learnable visual probe modules that interact layer-wise with intermediate feature representation via cross attention. It helps to extract domain-specific visual attributes without modifying original model. The method evaluated three diverse domain adaptation scenarios and perform competitively on the target domain while better retaining source domain performance.

**Strengths:**

- **Originality and Significance**: The method of using layer-wise visual probes to selectively extract new, domain-specific visual features from a frozen encoder is an excellent, novel approach that effectively decouples new learning from pre-trained knowledge, thereby successfully mitigating catastrophic forgetting.

- **Empirical Performance**: VISCOP demonstrates strong, consistent performance gains across all three complex domain adaptation tasks (cross-view, cross-modal, cross-task). It successfully achieves the best balance of target and source domains.

- **Interpretabilty**: VISCOP provided diagnostic studies including attention visualization, ablation studies which illustrate how the probes learns distinct and domain relevant representation that base VLM frozen envoder failed to capture.

**Weaknesses:**

- **Data selection**: In Table1-3, base VLM performa well in many of the target domain; This raises concern 1) potential overlap of source domain (and target domain) with the pretraining data of VLM 2) Substantial distribution shift might be less severe than claimed. Could authors explain, how these datasets selected?

**Questions:**

- Can the authors discuss the robustness of this optimal probe count and initialization across the diverse domain shifts?
- Does the optimal number of probes change significantly based on the severity of the distribution shift?

---

> ### Author Response · Authors · 2025-11-22
> **Response to Reviewer z5Ck (part 1)**
>
> We sincerely thank the reviewer for recognizing the critical novelty of our visual probing approach and for their positive assessment of VisCoP's effectiveness across three complex domain adaptation tasks. We are pleased that the reviewer found that our diagnostic studies and interpretability analysis clearly demonstrated that the visual probes learn domain-specific visual features that the base VLM failed to capture. We have carefully considered all points and provide detailed responses to the questions and concerns raised by the reviewer.
>
> ---
>
> >**Data selection: In Table 1-3, base VLM performs well in many of the target domain; This raises concern 1) potential overlap of source domain (and target domain) with the pretraining data of VLM 2) Substantial distribution shift might be less severe than claimed. Could authors explain, how these datasets selected?**
>
> * **Regarding data overlap concerns:** we clarify that the source domain refers to the distribution covered by the base VLM's pretraining data, so by definition there is alignment between them. However, we have confirmed that the target domains (EgoExo4D, VIMA-Bench) have no overlap with the pretraining data use in our base VLM (VideoLLaMA3), as these datasets are not listed in our base VLM's technical report [1]. **To address the reviewer’s concern and to enhance clarity, we have updated Section 5.2 in the paper with this discussion (blue font, E6)**
> * **Regarding lack of severe distribution shift:** the target domains were intentionally chosen to progressively increase in visual and instruction-level difficulty. The egocentric domain consists of RGB videos with a large viewpoint shift compared to primarily exocentric videos used to train the vase VLM. The depth setting replaces RGB entirely with depth, a visual modality the base VLM has never seen in its training. The robotics setting introduces both new embodiments (robot arms instead of humans) and new task instructions (taking actions rather than understanding videos) the base VLM has not seen during training. Because of the large gap between both visual content and instruction content of the chosen target domains and the source domain, we believe the chosen target domains represent three challenging domain-adaptation scenarios, as noted by reviewer `PRAm`. **The distribution shift between the source domain and the target domain becomes apparent when observing the performance of the base VLM on each target domain, where performance drops steadily as target domains shift further from the source domain.** We summarize Tables 1, 2, and 3 to present these performances in the table below, showing moderate degradation in the egocentric domain, a sharp degradation on the depth modality, and complete failure in the robotic control domain
>
> | Model&nbsp;&nbsp;&nbsp;    | Source Domain&nbsp;&nbsp;&nbsp; | Egocentric Target&nbsp;&nbsp;&nbsp; | Depth Target&nbsp;&nbsp;&nbsp; | Robot Control Target |
> |----------|--------------------------|------------------------------|--------------------------|---------|
> | Base VLM | 74.42                    | 70.43                        | 45.86                    | 0       |
>
> ---
>
> [1] Zhang et al., "VideoLLaMA 3: Frontier Multimodal Foundation Models for Image and Video Understanding", arxiv 2025

---

> ### Author Response · Authors · 2025-11-22
> **Response to Reviewer z5Ck (part 2)**
>
> >**Can the authors discuss the robustness of this optimal probe count and initialization across the diverse domain shifts?**
>
> Thank you for your insightful comment.
>
> * **Regarding the optimal visual probe count:** To address this concern, we conducted an additional analysis plotting the number of visual probes across all three target domains. We have updated the paper with these plots in Figure 5, along with a corresponding discussion in Section 5.3 (blue font, E8). In summary, we find that across all domains 16 probes consistently provides a reasonable tradeoff between target-domain gains and source-domain retention, with 16 probes being especially effective in the robotic control domain, which we discuss in more detail in Section 5.3
> * **To address the reviewer’s concern, we perform an additional experiment exploring two alternative initialization strategies for the visual probes:** zero-initialization (zero-init) and avg vision encoder feature (Avg VE) initialization. In zero-init, we initialize all of the visual probes with zeros. In Avg VE, we initialize the probes with the average pooled visual feature across all training samples. We present the results below, comparing with our default strategy of initialization from a normal distribution (normal init) on the egocentric target domain. We find that normal initialization provides the best balance, yielding strong target-domain gains while retaining more source-domain performance, suggesting that learning probe representations from scratch is more effective than anchoring them to fixed or averaged encoder features
>
> | Probe Init Strategy      | $\Delta_{target}$ | $\Delta_{source}$ |
> |--------------------------|-------------------|-------------------|
> | Zero init                | +3.39             | +0.79             |
> | Avg VE init              | +1.58             | +1.25             |
> | Normal init (default)    | +3.53             | +1.77             |

---

### Official Review · Reviewer_QbYg · 2025-11-01

**Soundness:** 3
**Presentation:** 4
**Contribution:** 4
**Rating:** 6
**Confidence:** 5

**Summary:**

This paper introduces VisCoP, a method for adapting Vision-Language Models (VLMs) to new video domains without catastrophic forgetting. In particular, from popular exocentric videos to egocentric videos and even for robot control and other visual modality. Then, the paper introduces a small set of learnable visual probes, which is used to learn domain-specific features, and preserve existing knowledges. In detail, without changing the output of pre-trained backbone, both the output from a probing network (cross attention) that corresponding to the probes are concatenated and fed into LLM together for downstream application, which seems pretty intuitive in method design. But still, the application task is indeed very important and challenging.

**Strengths:**

- The cross-domain application task for video analysis is very important, where the three application scenarios, particular for robot control, are comprehensive.
- Comprehensive evaluation and ablation studies on three domain transfer tasks.
- Both qualitative and quantitative analysis are provided.

**Weaknesses:**

- Despite comprehensive ablation studied, It looks like that the proposed method does not compare the other baselines clearly. As such, more comparison with (including repurposing any other methods, or methods that only focus on limited field transfer) is needed.
- Meanwhile, compared with prompting-based methods, the proposed methods clearly introduces more parameters and requires much more computation resource. However, a detailed explanation is still needed to let the audience check whether the surge in computation is still within a reasonable range.
- The novelty of the method design is limited. The proposed method has been studied in the image or language fields. For generalization over videos, I was expecting some video-specific techniques that can help with proposed evaluation setup, either in form of training objective or network design.
- Followingly, how do you guarantee that the outputs of probe will be exactly the domain-specific information for the new domains. How to you distinguish your methods from another baseline that just augments the network with more parameters? After all, I didn’t see clear optimization objective for the motivation mentioned in the introduction.

**Questions:**

Just out of curiosity, For line 402-403, “the resulting in 0% accuracy across all levels of VIMA-Bench” seems surprising. More explanation on the performance, metric definition is needed.

Please see my comments in weakness.

---

> ### Author Response · Authors · 2025-11-22
> **Response to Reviewer QbYg (part 1)**
>
> We thank the reviewer for recognizing the importance of the cross-domain adaptation problem we address and for their positive feedback on the comprehensiveness of both our target domain coverage and our analysis of what VisCoP learns. We have carefully considered all points raised by the reviwer and provide detailed responses below.
>
> ---
>
> >**Despite comprehensive ablation studied, It looks like that the proposed method does not compare the other baselines clearly. As such, more comparison with (including repurposing any other methods, or methods that only focus on limited field transfer) is needed.**
>
> Thank you for your concern. **To address this, we have incorporated a recently proposed domain adaptation approach into our comparisons, Model Tailor [1]**. Although Model Tailor was originally developed for LLaVA-1.5, which leverages the CLIP vision encoder and Vicuna LLM, we adapted the method to our setting, which leverages a SigLIP vision encoder and Qwen2.5 LLM to ensure a fair comparison. We add the results of Model Tailor to Table 5, with a corresponding discussion in Section 5.3 (blue font, E9). In summary, we find that VisCoP outperforms Model Tailor, which we hypothesize stems from VisCoP's ability to leverage intermediate vision encoder representations through visual probing, whereas Model Tailor only operates on the LLM layers, limiting its access to domain-specific visual features. We also note that the ablation strategies shown in Tables 1, 2, 3, and 4 represent most existing works that adapt VLMs to novel domains: through training only certain components of the base VLM on domain-specific instruction data [2].
>
> ---
>
> >**Meanwhile, compared with prompting-based methods, the proposed methods clearly introduces more parameters and requires much more computation resource. However, a detailed explanation is still needed to let the audience check whether the surge in computation is still within a reasonable range.**
>
> Thank you for your concern. **To address this, we have added Table 6 and updated Section 5.3 of the paper to quantify and discuss the computation overhead introduced by our proposed method (blue font, E2)**. In summary, we find that VisCoP introduces only 2% more parameters to the base VLM, and only increases visual feature inference latency by 0.013s.
>
> ---
>
> >**The novelty of the method design is limited. The proposed method has been studied in the image or language fields. For generalization over videos, I was expecting some video-specific techniques that can help with proposed evaluation setup, either in form of training objective or network design.**
> * We respectfully note that while domain adaptation has been explored in other fields, to the best of our knowledge, VisCoP is the first method to address this problem using visual probes that learn to extract domain-specific visual features from intermediate vision encoder representations, as noted by reviewers `z5Ck` and `PRAm`. Existing methods to domain adaptation in VLMs freeze the vision encoder entirely [2], preventing domain-specific visual features from being learned.
> * Regarding video-specific techniques, we posit that the interaction module in VisCoP is video specific, as the interaction module attends the visual probes to all video frames to learn spatio-temporal representations. This is in contrast to the vision encoder in existing VLMs, which only performs intra-frame attention [3, 4]. **To address the reviewer’s concern and to more adequately highlight this in the main paper, we have updated Section 4.2 with additional discussion (blue font, E3)**
>
> ---
>
> [1] Zhu et al., "Model Tailor: Mitigating Catastrophic Forgetting in Multi-modal Large Language Models", ICML 2024
>
> [2] Cheng et al., “On Domain-Adaptive Post-Training for Multimodal Large Language Models”, EMNLP Findings 2025
>
> [3] Bai et al., "Qwen2.5-VL Technical Report", arxiv 2025
>
> [4] Zhang et al., "VideoLLaMA 3: Frontier Multimodal Foundation Models for Image and Video Understanding", arxiv 2025

---

> ### Author Response · Authors · 2025-11-22
> **Response to Reviewer QbYg (part 2)**
>
> >**Followingly, how do you guarantee that the outputs of probe will be exactly the domain-specific information for the new domains. How to you distinguish your methods from another baseline that just augments the network with more parameters? After all, I didn’t see clear optimization objective for the motivation mentioned in the introduction.**
>
> * Thank you for your insightful question. Because the video-instruction pairs are implicitly domain-specific, our expectation is that the visual probes must implicitly learn to extract domain-specific information in order to minimize the auto-regressive training objective. **We qualitatively validate this through the attention visualization in Figure 6a, in which the visual probes implicitly learn to attend to the spatial regions relevant to the domain (e.g., the countertop where the action is occurring)**
> * **To address the reviewer’s concern that improvements from VisCoP may stem from the increased number of parameters, we present results of an additional experiment on the below, in which we inflate the vision-language connector such that the parameter count of the additional layers consists of the same number of parameters as VisCoP** (initializing them with the weights of the pre-trained VL-C layer). We find that simply adding additional parameters can actually hurt performance, which we hypothesize is because the LLM is forced to re-align itself to handle inputs from these additional layers, whereas VisCoP does not disrupt the pre-trained alignment of the VLM.
>
> | Adaptation Strategy        | $\Delta_{target}$ | $\Delta_{source}$ |
> |----------------------------|----------|----------|
> | Additional VL-C Parameters | -3.09    | -5.68    |
> | VisCoP                     | +3.53    | +1.77    |
>
> ---
>
> >**Just out of curiosity, For line 402-403, “the resulting in 0% accuracy across all levels of VIMA-Bench” seems surprising. More explanation on the performance, metric definition is needed.**
>
> Thank you for raising this question. In the robotic control target domain, accuracy corresponds to the success rate on the evaluated robotic manipulation tasks. The 0% success rate on these tasks occurs because the pre-training distribution of the base VLM lacks action trajectories (i.e., instruction data mapping from visual observations to robot actions), thus the base VLM has no ability to perform these robotic manipulation tasks, resulting in the 0% success rate seen in Table 3. Additionally, this finding is consistent with prior works such as LLaRA [1]. **To improve clarity, we have updated Section 5.2.3 in the paper with this discussion (blue font, E5).**
>
> ---
>
> [1] Li et al., “LLaRA: Supercharging Robot Learning Data for Vision-Language Policy”, ICLR 2025

---

### Official Review · Reviewer_gwox · 2025-11-01

**Soundness:** 3
**Presentation:** 3
**Contribution:** 2
**Rating:** 4
**Confidence:** 4

**Summary:**

This paper proposes a finetune method for VLM  when trying to improve VLM on the target domain. The proposed module VisCop share a similar design spirit with Q-former, which uses a few queries to probe the features.

**Strengths:**

Strength:

- This paper is well-written and easy to follow. The motivation and problem setting are clear.

- On the evaluation benchmarks, the proposed method shows general effectiveness.

**Weaknesses:**

Questions:

- Does $Acc_{source}^{expert}$ indicate the performance of the model after being finetuned on target domain data?

- In table 1, are Egocentric Benchmarks and Exocentric Benchmarks all target domain benchmarks?

- In table 1, why only the third method suffers from performance degeneration on the source domain? In addition, does "✓" mean fully tuning? I am curious about the root cause of source performance degeneration on the source domain. Is it fully fine-tuning LLM on the target domain?

- In table 2, Lora-tune the LLM seems to be a bad choice since it causes about 10$%$ of performance drop. So why not just train the model with only Viscop?

- When testing the process model on the source domain, will it process the source features with VisCop?

- VisCop introduces residual vision features, so what is the performance of using a LORA to introduce vision features?


Weakness:

- The technical contribution of this paper is not significant. The experiment design could be clearer to show more information. For example, to isolate the design choice that causes severe source performance generation and target performance gain.

**Questions:**

-

---

> ### Author Response · Authors · 2025-11-22
> **Response to Reviewer gwox (part 1)**
>
> We sincerely thank the reviewer for their recognition of our proposed approaches effectiveness across the evaluation benchmarks, and for finding that the problem setting and motivation of the work were clearly communicated. We have carefully considered all points and provide detailed responses to the questions and weaknesses raised by the reviewer.
>
> ---
>
> >**Does Acc[expert, source] indicate the performance of the model after being finetuned on target domain data?**
>
> Yes, in all cases other than “Base VLM”, the reported accuracies are the performance of the VLM after being finetuned on the target domain. “Base VLM” corresponds to the VLM trained only on the source domain, without being tuned on the target domain.
>
> ---
>
> >**In table 1, are Egocentric Benchmarks and Exocentric Benchmarks all target domain benchmarks?**
>
> In Table 1, only “Egocentric Benchmarks” correspond to the target domain benchmarks, “Exocentric Benchmarks” correspond to the source domain benchmarks. **To improve clarity, we have updated the headers of Tables 1, 2, 3, and 4 to more clearly reflect this correspondence.**
>
> ---
>
> >**In table 1, why only the third method suffers from performance degeneration on the source domain?**
>
> We attribute this performance degradation to the large number of trainable parameters when the VL-C, VE, and the full LLM are tuned, leading to overfitting. As for the improvements of other methods on the source domain, we attribute this to overlap between the language semantics of the source domain activities (egocentric ADL videos) and target domain activities in ADL-X (exocentric ADL videos). A more detailed reasoning is available on lines 340-343 of the paper.
>
> ---
>
> >**In addition, does "✓" mean fully tuning? I am curious about the root cause of source performance degeneration on the source domain. Is it fully fine-tuning LLM on the target domain?**
>
> Yes, a "✓" indicates that all parameters of the module are fully trainable, while entries like “LoRA” indicate the module is trained with LoRA adapters. As for the root of performance degradation on the source domain, we attribute this to catastrophic forgetting induced by full-parameter tuning on the target domain. Adapting highly-parameterized components such as the vision encoder enables domain-specific specialization, at the cost of forgetting pretrained representations learned from the source domain (line 51).
>
> ---
>
> >**In table 2, Lora-tune the LLM seems to be a bad choice since it causes about 10% of performance drop. So why not just train the model with only Viscop?**
>
> Thank you for this insightful suggestion. **To probe this idea, we conducted an additional experiment on the depth modality target domain in which we trained a VLM using only the vision-language connector (VL-C) and VisCoP, excluding LoRA tuning of the LLM.** We compared this model to our default training strategy, which trains the VL-C, VisCoP, and the LLM with LoRA. We present the results below, finding that the VL-C and VisCoP model is able to achieve large improvements on the target domain; however, our default strategy combining VisCoP with LoRA still performs best overall, which we attribute to LoRA enabling the LLM to better integrate the visual probes into its reasoning process.
>
> | Trainable components                 | $\Delta_{target}$ | $\Delta_{source}$  |
> |--------------------------------------|----------|-----------|
> | VL-C + VisCoP                        | +13.88   | -5.0      |
> | VL-C + VisCoP + LLM (LoRA)           | +19.27   | +1.84     |

---

> ### Author Response · Authors · 2025-11-22
> **Response to Reviewer gwox (part 2)**
>
> >**When testing the process model on the source domain, will it process the source features with VisCop?**
>
> Yes, even when processing the source domain the VisCoP module is used. **To improve clarity, we have updated the paper and noted this in Section 4.2.**
>
> ---
>
> >**VisCop introduces residual vision features, so what is the performance of using a LORA to introduce vision features?**
>
> Thank you for raising this concern. This ablation was conducted in Table 5 of the initial paper (where both the vision encoder and LLM are trained with LoRA), and found that training the vision encoder with LoRA is an ineffective strategy for domain adaptation in VLMs. A more detailed discussion of these results is available on lines 429-431.
>
> ---
>
> >**The technical contribution of this paper is not significant. The experiment design could be clearer to show more information. For example, to isolate the design choice that causes severe source performance generation and target performance gain.**
>
> * **Regarding technical contribution:** to the best of our knowledge VisCoP is the only method designed to extract domain-specific representations from intermediate representations of a frozen vision encoder using learnable visual probing tokens, as pointed out by reviewers `z5Ck` (who notes the “Originality and significance” of our contribution) and `PRAm` (who notes that “VisCoP introduces a novel, parameter-efficient mechanism for domain adaptation in VLMs”). In contrast, existing domain adaptation strategies for VLMs do not modify the VLM architecture, and instead fine-tune existing VLM components on domain-specific data [1] (lines 48-55, and lines 110-122).
>
> * **Regarding experimental design to isolate why domain adaptation is challenging for existing VLMs:** Our experiments show that this challenge stems from the necessity of training the vision encoder to effectively learn domain-specific representations. This is validated through the results of Tables 1, 2, and 3, where Vision Encoder training consistently leads to large performance gains on the target domain at the cost of performance loss on the source domain. **To address the reviewer’s comment and to make this observation more clear in the paper, we have updated Section 5.2.1 with this discussion (blue font, E1).**
>
> ---
>
> [1] Cheng et al., “On Domain-Adaptive Post-Training for Multimodal Large Language Models”, EMNLP Findings 2025

---

### Author Response · Authors · 2025-11-22
**Summary of key additions**

We sincerely thank all reviewers for their constructive feedback. To address the concerns raised by the reviewers, we have conducted additional experiments and have updated the main paper. The key additions include:

1. [Table 5] Comparison with Model Tailor, a recent domain adaptation method for VLMs.
2. [Table 5] Ablation on alternative interaction module architectures (self-attention vs. cross-attention).
3. [Table 6] Analysis of computational overhead (parameter count, VRAM, and inference latency).
4. [Figure 5] Probe count analysis across all three target domains.
5. [Section 5.3, E3] Discussion of video-specific design choices made in VisCoP's interaction module.
6. [Section 5.2.3, E5] Clarification on the 0% baseline accuracy in robotic control tasks.
7. [Section 5.2, E6] Clarification that there is no overlap between target domain data and base VLM pretraining data.
8. [Included in rebuttal responses] Additional analysis on trainable components and probe initialization strategies.
9. [Included in rebuttal responses] Comparison with fine-tuned VLM matching the number of parameters in VisCoP.

We have carefully considered the feedback given by each reviewer, and address their concerns individually below.

---

### Author Response · Authors · 2025-12-03
**Summary of our responses in discussion phase**

We would once again like to thank the reviewers for their valuable questions and feedback, and would also like to sincerely thank the AC for their effort spent reviewing the paper and discussions.

We were happy to see that three of the four reviewers recommended acceptance, and encouraged that reviewers found our work novel and significant (`z5Ck`, `PRAm`), effective (`gwox`, `z5Ck`, `PRAm`), backed by comprehensive analysis of the representations learned by VisCoP (`QbYg`, `z5Ck`, `PRAm`), and evaluated across challenging target domains (`QbYg`, `PRAm`).

We have carefully considered the concerns of all reviewers and provided detailed, point-by-point individualized responses along with additional experiments and analyses on November 22, 2025. We have also incorporated the key improvements we made in the discussion phase into the revised manuscript. As the discussion phase has concluded prematurely, below we provide the AC a fisheye view of our rebuttal, summarizing the primary concerns raised by the reviewers and the steps we have taken to address them.


---

>**Is LoRA tuning necessary for VisCoP's performance gains?**

We conducted an additional experiment training VisCoP without LoRA tuning in the LLM, demonstrating that LoRA is indeed necessary for the LLM to effectively leverage the visual probes.

---

>**How does VisCoP compare to other recent VLM domain adaptation methods?**

We incorporated Model Tailor [1], a recent VLM domain adaptation baseline, into our experiments (Table 5, Section 5.3) and demonstrated that VisCoP achieves superior performance.

[1] Zhu et al., "Model Tailor: Mitigating Catastrophic Forgetting in Multi-modal Large Language Models", ICML 2024

---

>**What is the computational overhead of VisCoP?**

We provided a comprehensive analysis (Table 6, Section 5.3) quantifying VisCoP's minimal overhead: 2% parameter increase, ~3GB additional VRAM usage, and 0.013s inference latency increase.

---

>**How many visual probes should be used, and how should they be initialized?**

We analyzed optimal probe counts across all target domains (Figure 5, Section 5.3), finding 16 probes provides the best tradeoff. We also evaluated multiple initialization strategies, demonstrating that normal initialization provides the best balance for both target and source domain performance.

---

>**What makes VisCoP's design specifically suitable for video domains?**

We expanded our discussion (Section 4.2) to clarify how VisCoP's interaction module attends across all video frames for spatio-temporal learning, distinguishing it from standard vision encoders used in VLMs for video understanding.

---

We believe that we have clarified the concerns and questions raised by the reviewers. However, if there are any additional thoughts or follow-up questions and if ICLR’s updated policies permit further discussion, we are more than happy to respond.

---

### Meta-Review · Area_Chair_PrHJ · 2025-12-28

**Summary:**

The reviewers’ concerns on limited novelty, unclear video-specificity, and weak mechanistic evidence, sadly led the AC’s recommendation to **reject**. Reviewer **QbYg** and **gwox** both questioned the conceptual novelty, which is closely aligned with established query/prompt-style mechanisms, yet the paper does not clearly position or distinguish VisCoP in that broader literature. Furthremore, reviewer **QbYg** explicitly expected more video-specific technique/justification beyond a generic probing module, as it is not clear why the method is considered a video-specific solution and not a broad image-based solution. Reviewer **QbYg** also raised that the claim that probes extract “domain-specific” information is not convincingly supported (visualizations are suggestive but do not rule out the alternative explanation that gains come largely from added capacity, and no explicit objective or sharper test is provided). Finally, while Table 5 adds helpful ablations, the baseline coverage remains incomplete for the alternatives (notably prompt-/token-based tuning inside the vision encoder).

**Reviewer Concerns:**

**Reviewer gwox:**
Authors clarified several issues (eg., table semantics), they also added an ablation showing VisCoP without LLM-LoRA underperforms the proposed algorithm. However, the reviewer’s core critique on limited technical contribution and the request to more cleanly isolate which design choices drive target gains vs source drops remains only partially addressed.

**Reviewer QbYg:**
Several issues such as adding a new baseline Model Tailor, compute/overhead analysis (params/VRAM/latency) are added. However, the concern regarding novelty was not fully addressed. Also the justification on “probes learning domain-specific information” is mostly qualitative/indirect (visualizations).

**Reviewer z5Ck:**
Authors responded to overlap/shift-severity and hyperparameter ablation concerns. No outstanding issues.

**Reviewer PRAm:**
Authors added missing ablations (probe initialization; alternative interaction module via self-attention vs cross-attention) and added compute overhead reporting. The concern that the method’s core mechanism is not clearly video-specific remains only partially answered.

**Reviewer Scores:**

The AC expects reviewers **gwox**, **z5Ck**, and **PRAm** to likely keep their original scores. For reviewer **QbYg**, it is plausible they would downgrade from $6 \to 4$, as their central objections (e.g., limited novelty and unclear video-specificity) appear only partially resolved in the rebuttal.

---

### Decision · Program_Chairs · 2026-01-26

Reject